# GranViT: A Fine-Grained Vision Model For Autoregressive Multimodal Large Language Models

**Guanghao Zheng[1][†], Bowen Shi[1][†], Mingxing Xu[2], Ruoyu Sun[2], Peisen Zhao[2],**
**Zhibo Zhang[2], Wenrui Dai[1][*], Junni Zou[1], Hongkai Xiong[1], Xiaopeng Zhang[2][*], Qi Tian[2]**
[1]Shanghai Jiao Tong University, Shanghai, China
[2]Huawei Inc., China

## Abstract

Vision encoders are indispensable for allowing impressive performance of Multimodal Large Language Models (MLLMs) in vision–language tasks such as visual question answering and reasoning. However, existing vision encoders focus on global image representations but overlook fine-grained regional analysis. They are limited in fine-grained perception due to the scarcity of fine-grained annotated data and the lack of a fine-grained pre-training paradigm. In this paper, we propose GranViT, a novel Vision Transformer that integrates fine-grained feature extraction with semantic alignment to Large Language Models (LLMs) via region-level autoregressive training. We first construct *Gran-29M*, a dataset comprising 29 million natural and OCR images paired with over 180 million high-quality region-level annotations, to enable large-scale fine-grained pretraining. Consequently, we develop a pretraining-adaptation framework along with a self-distillation mechanism to train fine-grained *GranViT* on *Gran-29M*. We sufficiently exploit the fine-grained annotations from *Gran-29M* to resort to bounding-box-to-caption regression to enhance localized visual representation of the vision encoder in the pretraining and caption-to-bounding-box regression to improve vision feature utilization and localization for LLM in the adaptation. We further incorporate a self-distillation mechanism that imposes explicit localization constraints on the vision encoder to strengthen its regional reasoning capability. Extensive experiments show that GranViT surpasses existing vision encoders and attains strong transferability to varying LLMs. Remarkably, it achieves state-of-the-art results on fine-grained recognition, multimodal VQA, and OCR understanding.

## 1 Introduction

Multimodal Large Language Models (MLLMs) have stimulated substantially growing research interests and efforts in recent years (Wang et al., 2024; Bai et al., 2025; Dong et al., 2025; Zhu et al., 2025; Wang et al., 2025; Li et al., 2025c). Existing architectures for MLLMs usually consist of a pretrained vision encoder that extracts visual information and a projection module that maps visual information to visual tokens for image understanding and reasoning with Large Language Models (LLMs). Projection modules such as multilayer perceptrons (MLPs) (Liu et al., 2023b) or Q-Formers (Li et al., 2023b) bridge visual features to the semantic space of LLMs, whereas vision encoders are primarily for the ability of capturing visual information for MLLMs.

Vision Transformers (ViTs) (Dosovitskiy et al., 2021) and their variants (Liu et al., 2021; Ravi et al., 2025; Zheng et al., 2024a) have been widely adopted as vision encoders in MLLMs due to their exceptional capabilities in visual feature extraction and scalability (Dosovitskiy et al., 2021; Carion et al., 2020; Kirillov et al., 2023; Ravi et al., 2025; Li et al., 2025b). Existing ViTs are usually trained to align visual representations with textual semantics. Contrastive Language–Image Pretraining (CLIP) (Radford et al., 2021; Zhai et al., 2023; Tschannen et al., 2025; Shi et al., 2024) is one prevailing paradigm that projects images and texts into a learned shared embedding space

---

[*]Correspondence to Wenrui Dai and Xiaopeng Zhang. [†] Equal contribution.

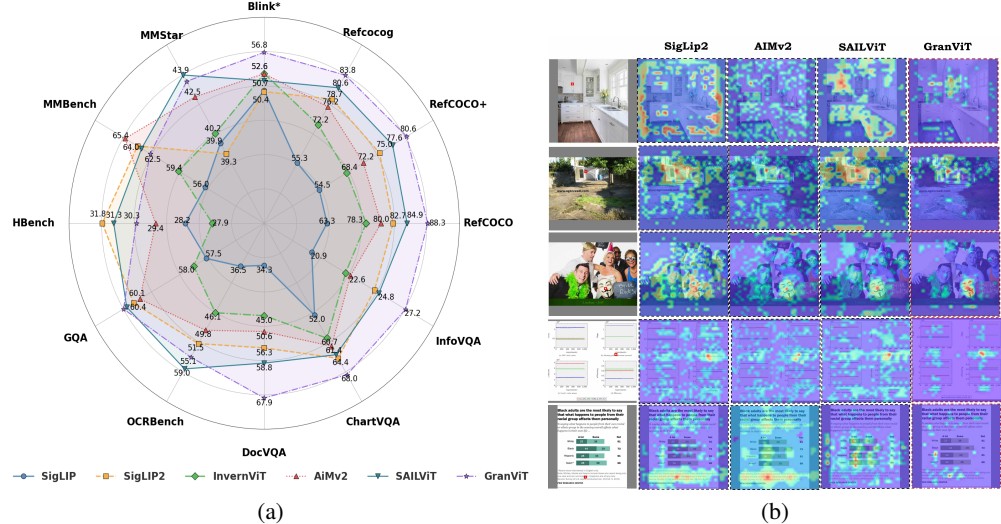

Figure 1: (a) Compared to existing vision encoders, GranViT demonstrates outstanding performance across fine-grained natural image and OCR understanding. HBench denotes HallusionBench. (b) Attention visualization of existing vision encoders according to the query token. The small red rectangle indicates the query token. Best viewed with zoom in.

to aggregate matched image-text pairs, and non-matching pairs are discriminated to preserve the semantic relationship. Another popular alternative is autoregressive modeling (Chen et al., 2024c; Fini et al., 2025; Tschannen et al., 2025) that directly maps visual features into the textual space by connecting a cascaded vision encoder and textual decoder. It allows superior alignment with textual space but could sacrifice the discrimination ability of visual features. Nevertheless, these approaches over-emphasize image-level global feature extraction but neglect essential fine-grained details required for multimodal understanding.

To address the limitation, in this paper, we investigate integrating fine-grained localization capabilities into the vision encoder within an LLM cascade architecture. It is non-trivial to address the following two challenges, *i.e.*, i) Data scarcity: scarcity of high-quality datasets with fine-grained annotations, and ii) Fine-grained pre-training: lack of a dedicated framework to train fine-grained vision encoders that effectively align with LLMs.

**i) Data scarcity.** We build a high-quality annotated dataset termed *Gran-29M* that contains 29 million natural and OCR images with image-level annotations along with 183 million region-level annotations. Specifically, we leverage the UMG-41M (Shi et al., 2024), FLICKR30k (Young et al., 2014), and LAION (Schuhmann et al., 2022) datasets to collect natural images of varying scales and diversity and generate image-level and region-level annotations (e.g., bounding boxes) using ViTDet (Li et al., 2022) and Qwen2.5-VL (Bai et al., 2025). Moreover, we consolidate publicly available OCR datasets (Li et al., 2025c; 2024a) and utilize PaddleOCR (Cui et al., 2025) for localized text detection and bounding box prediction. *Gran-29M* is achieved with rigorous filtering based on bounding box aspect ratio, area, quantity, and image resolution.

**ii) Fine-grained pre-training and adaptation.** We propose a novel pretraining-adaptation framework to improve fine-grained understanding of natural and OCR images beyond enhancing overall perception. In the pretraining, the proposed framework optimizes the vision encoder with bounding-box-to-caption ($Bbox2Caption$) regression for fine-grained feature extraction. Additionally, we develop localized self-distillation to optimize the vision encoder and explicitly augment its ability to extract fine-grained features. As for adaptation, the LLM is tunable for fine-grained vision feature localization with caption-to-bounding-box ($Caption2Bbox$) regression.

To validate the effectiveness of GranViT, we perform comprehensive performance comparisons and extensive visualizations after downstream supervised fine-tuning (SFT) (Li et al., 2025a), including visual question answering, visual grounding, and OCR understanding. Fig. 1 shows that GranViT achieves state-of-the-art performance on multiple benchmarks and exhibits strong generalization capabilities. The contributions of this work are summarized as below.

• We establish *Gran-29M*, a large-scale pretraining dataset containing 29 million natural and OCR images with comprehensive global annotations and 183 million fine-grained captions.

• We propose a pretraining-adaptation framework that simultaneously enhances the fine-grained feature extraction ability of GranViT with $Bbox2Caption$ regression and localized self-distillation using explicit local region supervision and adapts to varying LLMs with stronger capacity for local region localization with $Caption2Bbox$ regression.

• We demonstrate the robustness and generalization ability of *GranViT* compared with existing vision encoders via comprehensive analysis. *GranViT* achieves state-of-the-art performance on visual grounding and OCR comprehension.

## 2 RELATED WORK

### 2.1 MULTIMODAL LARGE LANGUAGE MODELS

Multimodal large language models (MLLMs) (Wang et al., 2024; Bai et al., 2025; Dong et al., 2025; Chen et al., 2024c;b; Zhu et al., 2025; Wang et al., 2025; Team et al., 2025b; Li et al., 2025c; Lei et al., 2025b) attract wide attention for their potential in image understanding and reasoning. Building on the robust textual understanding and reasoning capabilities of large language models (LLMs) (Touvron et al., 2023; Yang et al., 2025; Guo et al., 2025a; Yang et al., 2025; Lei et al., 2025a), most existing MLLMs augment their functionality with a pretrained vision encoder to enable visual perception. These encoders are usually trained with contrastive learning (Radford et al., 2021; Zhai et al., 2023; Tschannen et al., 2025; Shi et al., 2024) and projectors commonly adopt a two-layer MLP architecture (Li et al., 2025a; Liu et al., 2023b), but pre-trained vision encoders cannot handle high-resolution inputs (Zhai et al., 2023). Early models (Wang et al., 2024; Gu et al., 2024) adopt an image tiling strategy (Chen et al., 2024c; Liu et al., 2023b; Lu et al., 2024a; Team et al., 2025b): high-resolution images are divided into patches, from which local features are extracted and aggregated. In comparison, newer MLLMs such as Qwen2.5-VL (Bai et al., 2025), Seed-VL1.5 (Guo et al., 2025b), and Kimi-VL (Team et al., 2025a) train vision encoders from scratch on diverse datasets and support native-resolution input (Bai et al., 2025) to mitigate performance loss from resolution reduction. Beyond architectural improvements, recent MLLMs increasingly focus on post-training strategies (Cheng et al., 2025; Gu et al., 2024; Li et al., 2025c). These emphasize large-scale, curated SFT datasets and leverage both SFT and reinforcement learning (Schulman et al., 2017; Shao et al., 2024; Zheng et al., 2024b; 2026; 2024a) to enhance task-specific capability.

### 2.2 VISION FOUNDATION MODELS

Vision encoders are a critical component for extracting and representing visual information to support multimodal reasoning in MLLMs. Existing MLLMs usually employ vision encoders pre-trained through contrastive learning, which inherently align visual and textual semantic spaces. Commonly used encoders include CLIP (Radford et al., 2021) using cross-entropy loss (Mao et al., 2023), and SigLIP (Zhai et al., 2023) using sigmoid loss. InternViT (Chen et al., 2024c) combines contrastive learning with an autoregressive loss and incorporates a text decoder to enhance alignment by decoding visual features into text. SeedViT (Guo et al., 2025b) first undergoes generative self-supervised pretraining (Xie et al., 2022; He et al., 2022) before contrastive learning. AIMv2 (Fini et al., 2025) introduces the first vision encoder trained solely with an autoregressive loss, predicting subsequent image patches and text tokens to achieve cross-modal alignment without contrastive learning. SigLIP2 (Tschannen et al., 2025) integrates autoregressive and self-distillation losses in SigLIP to enhance visual representations through multi-objective pretraining. SAILViT (Yin et al., 2025) extends AIMv2 by incorporating alignment with LLMs and multi-stage pretraining with SFT data, facilitating high-dimensional vision-language integration and infusing world knowledge into visual encoding. However, these encoders predominantly emphasize global feature extraction at the cost of fine-grained visual details, and are limited in fine-grained multimodal tasks.

## 3 GRAN-29M: FINE-GRAINED ANNOTATED DATASET

In this section, we elaborate on the construction of a large-scale fine-grained *Gran-29M* dataset for pre-training, including data sources, data annotations, filtering criteria, and data reformatting.

**Data Source.** We collect diverse large-scale images from public datasets. For natural images, we expand the UMG-41M dataset (Shi et al., 2024) (including CC3M (Sharma et al., 2018), IN21k (Deng

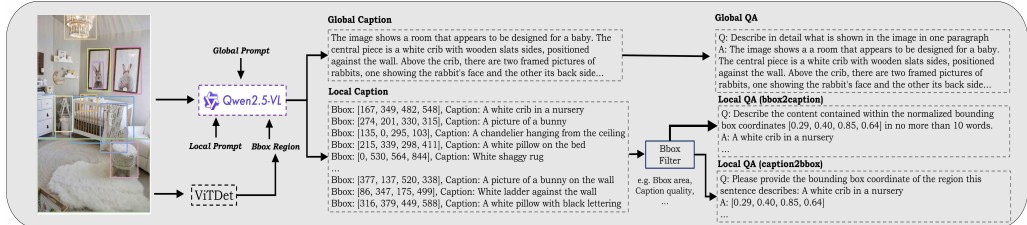

Figure 2: The details of data annotations of *Gran-29M*. We leverage ViTDet (Li et al., 2022) and Qwen2.5-VL-7B (Bai et al., 2025) for bbox and caption generation. Then, we transfer the absolute bbox coordinate to a relative one based on the image resolution and apply rigorous filtering based on image resolution, bbox area, and the number of bboxes per image. Finally, we reformat the global and local captions into QA pairs.

et al., 2009), SBU (Ordonez et al., 2011), CC12M (Changpinyo et al., 2021), YFCC15M (Kamath et al., 2021), and VisualGenome (Krishna et al., 2017)) with samples from LAION (Schuhmann et al., 2022) and FLICKR30k (Young et al., 2014). For OCR images, we collect four distinct types of images from publicly available sources (Li et al., 2024a; 2025c): plain text images, chart and table images, receipt images, and rich text images. Refer to Table 6 in the appendix for details.

**Data Annotations Workflow.** For natural images, we directly utilize bounding box (bbox) coordinates provided by UMG-41M as localized annotations for local regions, and employ Qwen2.5-VL-7B (Bai et al., 2025) to regenerate global and local captions to enhance caption quality. For the LAION dataset (Schuhmann et al., 2022) and FLICKR30k (Young et al., 2014), we utilize ViTDet (Li et al., 2022) to detect bboxes and Qwen2.5-VL-7B (Bai et al., 2025) to generate global and local captions, as shown in Fig. 2. For OCR images, since global descriptions are often vague (*e.g.*, "a page of an academic paper") and lack details, only local regions are annotated using PaddleOCR (Cui et al., 2025) to provide accurate bboxes and textual contents simultaneously.

**Filtering Criteria.** To ensure high-quality annotations for both global and local regions, we apply a filtering process based on image resolution and bbox criteria. For local region annotations, we require that the shorter side of each image should be larger than $448$ pixels, the aspect ratio of both the entire image and each bbox should be between $\frac{1}{3}$ and 3, the area of each bbox should be greater than $100^2$ square pixels, and the number of bboxes per image should be at least one. The filtered results are summarized in Table 7 in the appendix. In total, we obtain 29.51 million images with 183.55 million localized region annotations for large-scale pretraining.

**Data Reformatting.** To facilitate the training of GranViT, we reorganize existing global and local region captions and reformat them into a standard question-answer pair structure. Using the following question and answer prompts, we rewrite existing data to enhance its suitability for training. For bbox coordinates and corresponding captions, we perform bidirectional annotations through $Bbox2Caption$ and $Caption2Bbox$ tasks for the vision encoder and LLM pretraining, respectively. Furthermore, we convert the absolute bbox coordinates into relative coordinates based on image resolutions to eliminate the dependence on absolute coordinates.

**Global Caption.**
```
Q: Describe in detail what is shown in the image in one
paragraph
A: [global captions]
```

**Bbox2Caption.**
```
Q: Describe the content contained within the normalized
bounding box coordinates [bbox coordinates] in no more
than 10 words.
A: [local captions]
```

**Caption2Bbox.**
```
Q: Please provide the bounding box coordinate of the
region this sentence describes:  [local captions]
A: [bbox coordinations]
```

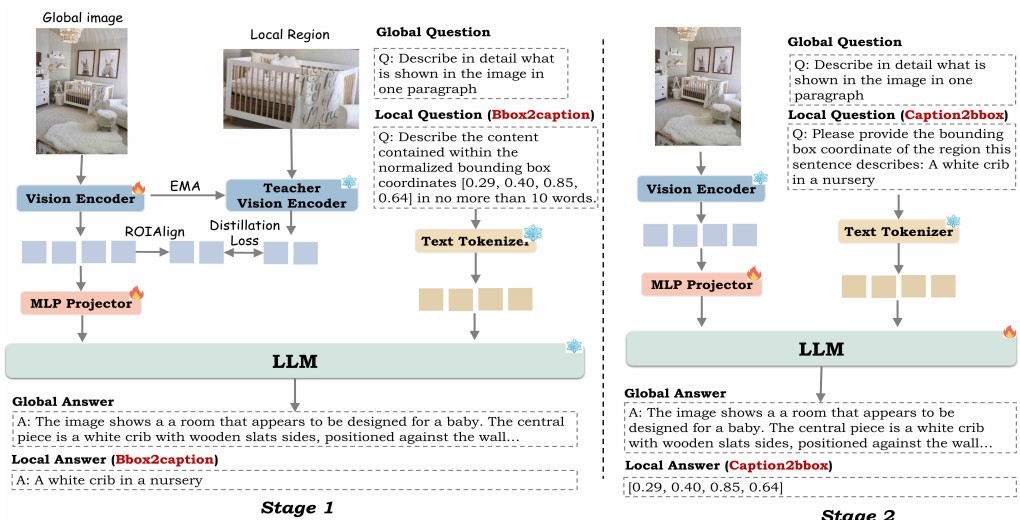

Figure 3: The fine-grained pretraining and transferring paradigm of GranViT. For pretraining, the vision encoder and projector are tuned via the global and $Bbox2Caption$ task for fine-grained feature extraction. The teacher vision encoder explicitly supervises the local region of features extracted by the student vision encoder. For vision feature adaptation and transfer, based on the fine-grained vision encoder, we apply LLM tuning to further strengthen the localization capability of the LLM regarding fine-grained visual features via the global and $Caption2Bbox$ task.

## 4 PROPOSED METHOD

### 4.1 FINE-GRAINED PRETRAINING PARADIGM WITH AUTOREGRESSIVE PERCEPTION

Owing to training solely on images and global captions (Radford et al., 2021; Zhai et al., 2023; Fini et al., 2025), previous vision encoders struggle with fine-grained feature extraction for local regions, while also lacking alignment between visual features and the textual feature of the LLM. To overcome these issues with a unified framework, we leverage the LLM to provide supervision for the fine-grained training of vision encoders. Specifically, we employ the same global image captioning task (Radford et al., 2021; Zhai et al., 2023; Fini et al., 2025) throughout the entire pretraining process to preserve the global perception capability. Furthermore, we enhance the ability of fine-grained feature extraction by cascading the vision encoder with the LLM via the projector during pretraining, and perform large-scale pretraining using both $Bbox2Caption$ and $Caption2Bbox$ tasks for localized region recognition and grounding, respectively. As depicted in Fig. 3, the proposed framework consists of pretraining and adaptation stages.

- **Stage 1: Pretraining that tunes vision encoder and projector with LLM frozen.** We additionally employ the $Bbox2Caption$ task for pretraining, which requires the MLLM to generate a localized caption of the object within specified bboxes. The LLM can be viewed as a decoder that converts visual features into texts, where the supervision is directly propagated back to the extracted local features with bboxes, thereby enhancing the fine-grained characteristics of the visual representations. The input prompt to the LLM incorporates the bbox coordinates, thereby facilitating object recognition and localization. It is worth noting that, in this stage, we employ a lightweight LLM (*i.e.*, Qwen2.5-VL-1.5B (Bai et al., 2025)) to compel the vision encoder to extract generic fine-grained features, rather than relying on the powerful reasoning capabilities of large LLMs for output generation.

- **Stage 2: Adaptation and Transfer that tune projector and LLM with the vision encoder frozen.** In contrast, we employ the $Caption2Bbox$ task in this stage, which requires the MLLM to recognize objects present in the image according to the prompts and output their bbox coordinates. The primary objectives of this stage are to further enhance the localization capability of the LLM based on fine-grained visual features and to ensure the transferability of the vision encoder to other LLMs with comparable or larger size. Since in MLLMs, it is required that the vision encoder provide more fine-grained features, while the LLM should also be capable of utilizing these visual features. On the other hand, the vision

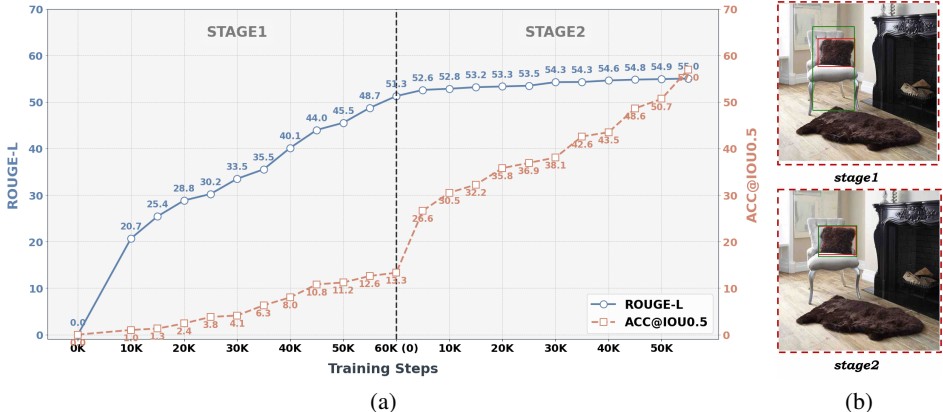

(a)                     (b)

Figure 4: (a) The performance curve of Stage1 and Stage2. We sample 8M $Bbox2Caption$ and $Caption2Bbox$ samples respectively for pretraining and adaptation and calculate ROUGE-L (Barbella & Tortora, 2022) and ACC@IOU0.5 for $Bbox2Caption$ and $Caption2Bbox$ respectively. In stage 1, the ACC@IOU0.5 of the $Caption2Bbox$ task only achieves $13\%$, while the ROUGE-L of the $Bbox2Caption$ task achieves $52\%$. Conversely, in stage 2, the training of LLM leads to a notable increase in ACC@IOU0.5 for $Caption2Bbox$, while $Bbox2Caption$ achieves only a minimal improvement of $3\%$. (b) Visualization of predicted bbox coordinate of $Caption2Bbox$ task in stage 1 and stage 2. Green bboxes indicate predicted regions, while red ones denote ground truth.

> encoder can be adapted with different LLMs (*i.e.*, Qwen2.5-VL-3B, Qwen2.5-VL-7B (Bai et al., 2025)), ensuring compatibility and transferability to new architectures.

Across both stages, the autoregressive caption loss is applied to regulate the text output of the LLM for supervising the vision encoder and the LLM, respectively. Given ground truth text $T$ and output text $O_{LLM}$, the caption loss can be calculated by $L_{caption} = CrossEntropy(O_{LLM}, T)$. Through this pretraining-adaptation pretraining paradigm, the vision encoder gains enhanced fine-grained feature extraction abilities, with its outputs inherently aligned to the semantic space of the LLM. The LLM, in turn, improves its capacity to utilize visual information for accurate localization. These generalized capabilities enhance object recognition and spatial reasoning during downstream SFT, thereby reducing visual understanding errors and mitigating hallucinations.

## 4.2 EXPLANATION OF FINE-GRAINED PRETRAINING PARADIGM

**Pretraining-Adaptation framework with different tasks.** To enhance the fine-grained feature extraction capability of the vision encoder, in pretraining, we freeze the LLM and only train the vision encoder and projector in stage 1. Both the $Bbox2Caption$ and $Caption2Bbox$ tasks are used initially; however, as illustrated in Fig. 4(a), the model shows strong performance in $Bbox2Caption$ but limited accuracy in $Caption2Bbox$, primarily due to the frozen LLM, which restricts learning for the more language-reliant $Caption2Bbox$. Thus, we further apply stage 2 to adapt the pretrained vision encoder to LLMs to utilize fine-grained visual features. Meanwhile, the pretrained vision encoder can be adapted with other LLMs for transferability. Specifically, in stage 2, we keep the vision encoder frozen and tune the projector and LLM using both $Bbox2Caption$ and $Caption2Bbox$ tasks. Results in Fig. 4(a) demonstrate a notable improvement in ACC@IOU0.5 for $Caption2Bbox$, while $Bbox2Caption$ sees only marginal gains. This finding is consistent with the visualization results in Fig. 4(b), confirming the superiority of training $Caption2Bbox$ in stage 2. Since further optimizing $Bbox2Caption$ in stage 2 yields minimal benefits at twice the computational cost, we finally adopted the pretraining-adaptation paradigm, separately optimizing the two tasks in these two stages.

**Freezing the vision encoder in the adaptation stage.** We freeze the vision encoder and only tune the LLM for adaptation and transfer in stage 2, enabling the LLM to better use the fine-grained vision feature for the $Caption2Bbox$ task. The vision encoder is kept frozen because it has already been sufficiently pretrained in stage 1, and further tuning it would significantly increase adaptation and transfer costs but yield marginal performance gains, as shown in Table 9 in the appendix.

### 4.3 SELF DISTILLATION FOR FINE-GRAINED PRETRAINING PARADIGM

Although the vision encoder is implicitly supervised through the caption loss $L_{caption}$ from the output text of LLM, it lacks explicit constraints on localized region features. Therefore, we incorporate self-distillation training (Naeem et al., 2024; Maninis et al., 2025; Zhang et al., 2019) into stage 1. As illustrated in Fig. 3, an additional frozen teacher vision encoder is introduced to constrain the feature generated by the student vision encoder, thereby enhancing localized region features.

Specifically, given images $x$, the student vision encoder extracts the fine-grained features $x'$. Both prompts and visual features $x'$ are sent to the LLM for generating localized captions. Besides, the image regions $x_{crop}$ are cropped from $x$ according to the bbox coordinates and are sent to the teacher vision encoder for localized feature extraction. The self-distillation loss is calculated between $x'$ and $x'_{crop}$ by $L_{distill} = MSE(x'_{crop}, ROIAlign(x'))$, where $x'_{crop}$ and $MSE$ denotes the extracted features of $x_{crop}$ and mean square error loss, respectively. The weights of the teacher vision encoder are initialized from the student vision encoder and updated by the exponential moving average (EMA) according to the student vision encoder, specifically $\theta_{tea} = \alpha\theta_{tea} + (1-\alpha)\theta_{stu}$. The overall loss can be written as $L = L_{caption} + \lambda L_{distill}$, where $\lambda$ denotes the weighting coefficient.

## 5 EXPERIMENTS

### 5.1 EXPERIMENTAL SETTINGS

**Dataset.** Three datasets are adopted for distinct purposes of projector pre-training, fine-grained pretraining, and downstream SFT. Following (Liu et al., 2023b), we adopt the BLIP-LAION-CC-SBU-558K dataset (Liu, 2024) for projector pretraining. For fine-grained pretraining, we employ the constructed *Gran-29M* dataset. To balance natural and OCR images, we proportionally sample 50M local region captions from OCR regions and use all the natural image local captions. During stage 1, we use all global samples and approximately 130M $Bbox2Caption$ samples. In stage 2, we sample 24M $Caption2Bbox$ samples in addition to global samples to reduce the transfer overhead. For SFT, we adopt the Open-LLaVA-NeXT 1M dataset (Chen & Xing, 2024) for downstream adaptation.

**Implementation Details.** We use the LLaVA-Next framework (Li et al., 2025a) for pre-training and SFT of the vision encoder and LLM. By default, we initialize the vision encoder with SigLIP2 (Tschannen et al., 2025), and adopt Qwen2.5-VL-1.5B (Bai et al., 2025) as the LLM. The projector is implemented with a two-layer MLP. In training, images are resized to $512 \times 512$, padded according to their aspect ratio, and then fed into the vision encoder and LLM for feature extraction and inference. We also implement GranViT with image tiling strategy, as shown in Table 8 in the appendix. We employ AdamW optimizer (Loshchilov & Hutter, 2019) for pretraining and SFT with learning rate $10^{-5}$ for one epoch. The overall batch size is set to 256 with 128 Ascend 910B NPUs. $\lambda$ is 1 and $\alpha$ is 0.9 by default in our experiment. Abation study on $\lambda$ and $\alpha$ is shown in Table 5 in the appendix.

### 5.2 BENCHMARK EVALUATION

We make extensive evaluations on the well-known OpenCompass benchmark (OpenCompass Contributors, 2023) and additional fine-grained benchmarks. We focus on fine-grained and OCR benchmarks that are divided into four classes: fine-grained (RefCOCO (Yu et al., 2016), RefCOCO+ (Yu et al., 2016), RefCOCOg (Yu et al., 2016), BLINK* (Fu et al., 2024)[1] and MMVP (Tong et al., 2024)), mulitmodal VQA (MMBench (Liu et al., 2024b), MMStar (Chen et al., 2024a), Hallusion-Bench (Guan et al., 2024), GQA (Hudson & Manning, 2019) and SEEDBench (Li et al., 2023a)), multimodal reasoning (MMMU (Yue et al., 2024), MathVista MINI (Lu et al., 2024b), MMVet (Yu et al., 2024), ScienceQA (Lu et al., 2022) and AI2D (Kembhavi et al., 2016)) and OCR understanding (OCRBench (Liu et al., 2024c), DocVQA (Mathew et al., 2021), ChartQA (Masry et al., 2022), InfoVQA (Mathew et al., 2022) and TextVQA (Singh et al., 2019)). We compare GranViT with diverse vision encoders, *i.e.*, CLIP (Radford et al., 2021), SigLip (Zhai et al., 2023), SigLip2 (Tschannen et al., 2025), AIMv2 (Fini et al., 2025), InternViT (Chen et al., 2024c), and SAILViT (Yin et al., 2025).

---

[1]We calculate the average score of fine-grained evaluation (Counting, Object Localization and Spatial Relation) in BLINK, denoted as BLINK*

Table 1: Performance comparison with low resolution version. The bold font represents the best performance, and the underline represents the second performance.

| Capability | Benchmark | CLIP | SigLip | SigLip2 | AIMv2 | InternViT | SAILViT | GranViT |
|---|---|---|---|---|---|---|---|---|
| | RefCOCO_testA | 81.26 | 69.47 | 87.78 | 86.03 | 85.15 | 89.65 | **91.79** |
| | RefCOCO_testB | 64.51 | 56.78 | 76.90 | 73.54 | 71.40 | 79.82 | **83.88** |
| | RefCOCO_val | 74.71 | 63.71 | 83.26 | 80.28 | 78.48 | 85.32 | **89.13** |
| | RefCOCO+_testA | 74.43 | 63.13 | 82.92 | 81.33 | 77.33 | 85.01 | **87.04** |
| | RefCOCO+_testB | 51.25 | 44.65 | 66.47 | 62.50 | 58.58 | 69.66 | **73.24** |
| Fine-Grained | RefCOCO+_val | 65.25 | 55.84 | 75.46 | 72.84 | 69.18 | 78.10 | **81.55** |
| | RefCOCOg_val | 68.95 | 55.02 | 78.53 | 76.55 | 71.62 | 80.26 | **83.86** |
| | RefCOCOg_test | 68.52 | 55.57 | 78.94 | 75.78 | 72.71 | 80.92 | **83.82** |
| | BLINK* | 51.87 | 50.67 | 50.35 | 52.59 | 52.62 | 52.54 | **56.80** |
| | MMVP | 63.33 | 61.66 | 66.00 | 65.66 | 61.00 | **69.00** | 66.33 |
| | Average | 66.41 | 57.67 | 75.61 | 73.50 | 70.53 | 77.95 | **80.78** |
| | MMBench | 61.14 | 55.95 | 64.00 | **65.40** | 59.44 | 63.54 | 62.46 |
| | MMStar | 39.46 | 39.93 | 39.33 | 42.53 | 40.20 | **43.86** | 43.73 |
| VQA | HallusionBench | 28.83 | 28.22 | **31.79** | 29.39 | 27.95 | 31.29 | 30.34 |
| | GQA | 58.80 | 57.52 | 60.42 | 60.14 | 58.02 | 60.83 | **60.95** |
| | SEEDBench | 66.89 | 66.28 | 69.30 | 70.20 | 66.79 | 69.75 | **70.36** |
| | Average | 51.02 | 49.58 | 52.97 | 53.53 | 50.48 | **53.85** | 53.57 |
| | MMMU | 40.00 | 38.66 | 42.00 | 38.66 | **44.00** | 38.66 | 38.00 |
| | MathVista MINI | 38.10 | 35.50 | 38.40 | 37.70 | 39.50 | **41.70** | 40.40 |
| Reasoning | MMVet | 33.53 | 30.87 | **38.80** | 38.34 | 35.27 | 35.73 | 37.29 |
| | ScienceQA | 67.87 | 66.53 | 66.98 | 67.57 | 66.63 | **72.78** | 67.42 |
| | AI2D | 66.51 | 65.47 | 69.26 | 68.19 | 66.51 | **71.21** | 69.91 |
| | Average | 49.20 | 47.41 | 51.09 | 50.09 | 50.38 | **52.02** | 50.60 |
| | OCRBench | 406 | 365 | 515 | 498 | 461 | **590** | 551 |
| | DocVQA | 35.34 | 34.26 | 56.32 | 50.56 | 44.95 | 58.75 | **67.92** |
| OCR | ChartQA | 50.84 | 51.96 | 64.44 | 61.36 | 60.68 | 63.24 | **67.96** |
| | InfoVQA | 20.67 | 20.89 | 24.12 | 22.60 | 21.91 | 24.75 | **27.19** |
| | TextVQA | 46.65 | 41.71 | 61.36 | 56.57 | 51.74 | 60.90 | **61.66** |
| | Average | 38.82 | 37.06 | 51.55 | 48.18 | 45.08 | 53.33 | **55.97** |

**Performance Comparison.** Table 1 provides a performance comparison on various benchmarks. For fine-grained and OCR tasks, GranViT achieves an average top-1 score of 80.78 and 55.97, and surpasses the second best by 2.83 and 2.64, respectively. GranViT is comparable to SAILViT with a marginal difference of only 0.3 in multimodal VQA tasks. For multimodal reasoning tasks, GranViT suffers a slight performance loss of 0.4 compared to SigLIP2. This is because reasoning capability does not rely heavily on fine-grained feature extraction, and GranViT prioritizes fine-grained tasks rather than extensively training reasoning capabilities. Note that reasoning capability can be further improved by applying reasoning VQA data in pretraining.

**Transferability.** Table 2 reports the performance comparison with larger LLMs (*i.e.*, Qwen2.5-3B, Qwen2.5-7B). Notably, other vision encoders directly employ the larger LLM for SFT, whereas GranViT employs a lightweight LLM (*i.e.*, Qwen2.5-1.5B) for pre-training in stage 1, transfers to the larger LLM in stage 2, and subsequently undergoes SFT. GranViT also demonstrates outstanding performance on fine-grained and OCR tasks, while achieving comparable or even state-of-the-art performance on some VQA tasks (*i.e.*, HallusionBench and SEEDBench).

## 5.3 SCALING LAWS

We evaluate the scaling capacity of pretraining-adaptation framework in Fig. 5. Specifically, for stage 1, we leverage 8M, 16M and all the 130M regions for $Bbox2Caption$ tasks, while we leverage 8M, 16M, 24M and all the 130M regions for $Caption2Bbox$ tasks in stage 2. The average score of fine-grained (RefCOCO_testA, RefCOCO+_testA, RefCOCOg_test, BLINK, MMVP) and OCR (OCRBench, DocVQA, ChartQA, InfoVQA, TextVQA) tasks is reported. As the data scale increases, both tasks exhibit significant performance improvements, indicating enhanced fine-grained feature extraction capability of GranViT.

Table 2: Performance comparison for transferring vision encoders to Qwen2.5-3B, Qwen2.5-7B and LLaMA3-8B. The best results are highlighted in bold and the second best underlined. Ref, Ref+ and Refg denote the RefCOCO_testA, RefCOCO+_testA and RefCOCOg_test. MMB, HB, and SB stand for MMBench, HallusionBench, and SEEDBench, and. SQA, OB, DVQA, and IVQA for ScienceQA, OCRBench, DocVQA, and InfoVQA, respectively.

| Model | Ref | Ref+ | Refg | MMB | HB | SB | MMMU | SQA | OB | DVQA | IVQA | Avg |
|---|---|---|---|---|---|---|---|---|---|---|---|---|
| Qwen2.5-3B | | | | | | | | | | | | |
| CLIP | 86.60 | 81.73 | 75.26 | 65.09 | 31.67 | 68.55 | 39.33 | 73.12 | 413 | 38.70 | 24.08 | 56.86 |
| SigLip | 87.55 | 82.83 | 77.46 | 69.27 | 33.12 | 70.29 | 36.44 | 70.64 | 428 | 46.27 | 25.24 | 58.36 |
| SigLip2 | 91.03 | 86.37 | 83.30 | 69.34 | 33.70 | 71.13 | 36.00 | 71.69 | 529 | 60.87 | 29.00 | 62.30 |
| AIMv2 | 90.33 | 87.05 | 81.83 | 69.27 | 31.91 | 71.84 | **40.00** | 74.46 | 545 | 56.04 | 27.34 | 62.23 |
| SAILViT | 91.58 | 87.82 | 83.63 | **69.73** | **34.08** | 71.33 | 36.00 | **75.75** | **633** | 62.53 | 29.49 | 64.11 |
| GranViT | **93.22** | **89.32** | **86.17** | 67.56 | 33.77 | **72.34** | 38.66 | 73.24 | 590 | **71.09** | **29.96** | **64.94** |
| Qwen2.5-7B | | | | | | | | | | | | |
| CLIP | 90.01 | 86.25 | 80.44 | 70.58 | 36.39 | 70.70 | **46.00** | 75.26 | 466 | 43.02 | 24.88 | 60.92 |
| SigLip | 90.68 | 86.27 | 81.72 | 74.22 | 37.94 | 72.04 | **46.00** | 76.99 | 459 | 50.03 | 26.69 | 62.59 |
| SigLip2 | 92.06 | 88.70 | 85.13 | 72.21 | 36.02 | 72.17 | 44.00 | 75.26 | 540 | 62.20 | 29.89 | 64.69 |
| AIMv2 | 90.84 | 87.82 | 82.39 | 72.29 | 37.29 | 72.11 | 41.44 | 72.92 | 553 | 56.54 | 28.59 | 63.41 |
| SAILViT | 92.66 | 89.50 | 85.32 | **74.53** | 37.76 | 73.05 | 44.66 | **81.11** | **648** | 64.13 | 30.66 | 67.11 |
| GranViT | **92.98** | **90.46** | **87.96** | 73.37 | **39.37** | **74.45** | 44.66 | 75.85 | 582 | **73.14** | **31.69** | **67.47** |
| LLaMA3-8B | | | | | | | | | | | | |
| CLIP | 90.26 | 86.43 | 80.59 | 72.07 | 37.65 | 72.18 | 48.00 | 74.90 | 462 | 44.28 | 24.14 | 61.38 |
| SigLip | 90.47 | 86.44 | 81.44 | 74.02 | 35.25 | 74.55 | 48.97 | 76.48 | 473 | 50.69 | 23.37 | 62.63 |
| SigLip2 | 92.85 | 88.59 | 85.59 | 72.31 | 37.99 | 74.36 | 45.62 | 76.04 | 562 | 64.72 | 27.56 | 65.62 |
| AIMv2 | 91.30 | 88.03 | 83.67 | 72.49 | 37.02 | 73.77 | 40.77 | 73.81 | 529 | 58.85 | 24.04 | 63.33 |
| SAILViT | 93.06 | 89.94 | 85.89 | **74.08** | 39.02 | 75.35 | 45.66 | **79.59** | **640** | 68.97 | 28.08 | 67.60 |
| GranViT | **93.65** | **91.27** | **88.23** | 73.08 | **39.92** | **76.37** | 47.14 | 78.42 | 625 | **77.24** | **31.44** | **69.02** |

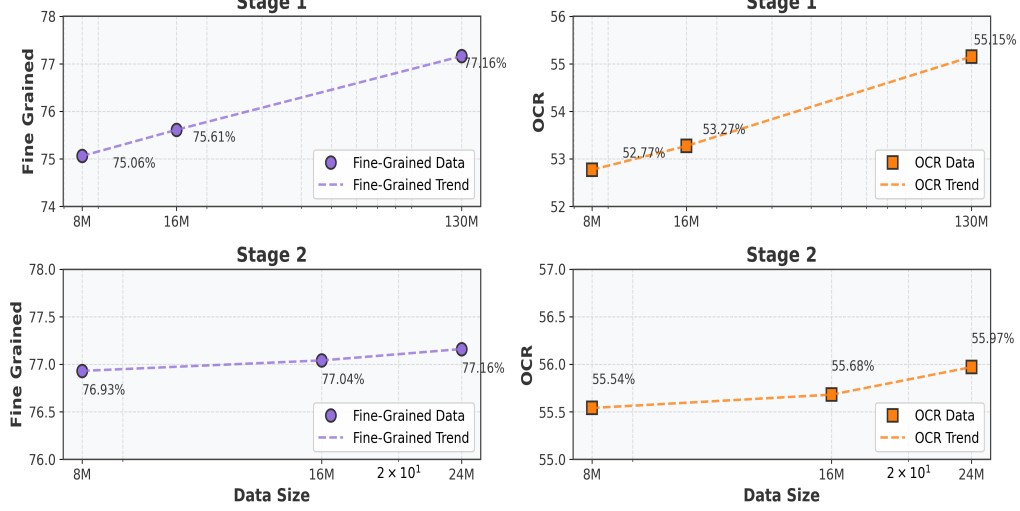

Figure 5: Scaling law of two-stage training.

## 5.4 ABLATION STUDY

We employ small-scale datasets for ablation studies on the contribution of each module. 8 million global and $Bbox2Caption$ QA pairs and 8 million global and $Caption2Bbox$ QA pairs are sampled from *Gran-29M* as training data for two stages, respectively. The entire dataset is used for training the projector and conducting supervised fine-tuning (SFT).

**Effectiveness of Training Paradigm.** In Table 3, the progressive introduction of the two-stage training strategy along with self-distillation results in incremental performance gains on fine-grained and OCR-related tasks. Specifically, with stage 1 pretraining, the MLLM exhibits a substantial improvement in both fine-grained recognition capability and OCR understanding (2.2 and 1.2 gains). Self-

Table 3: Ablation study on each component of the proposed GranViT.

| SigLip2 | Stage1 | Self-Distillation | Stage2 | Fine-Grained | VQA | Reasoning | OCR |
|---|---|---|---|---|---|---|---|
| ✓ | ✗ | ✗ | ✗ | 73.20 | 52.97 | **51.09** | 51.55 |
| ✓ | ✓ | ✗ | ✗ | 75.06 | 53.64 | 49.89 | 52.77 |
| ✓ | ✓ | ✓ | ✗ | 75.55 | **53.90** | 50.32 | 53.02 |
| ✓ | ✓ | ✓ | ✓ | **76.54** | 53.77 | 48.99 | **53.78** |

Table 4: Performance with different vision encoder initialization for GranViT during pretraining.

| Model | Fine-Grained | VQA | Reasoning | OCR |
|---|---|---|---|---|
| InternViT | 69.76 | 50.48 | **50.38** | 45.08 |
| GranViT (InternViT) | **75.15** | **51.78** | 50.23 | **50.13** |
| AIMv2 | 72.28 | 53.53 | 50.09 | 48.18 |
| GranViT (AIMv2) | **77.14** | **55.07** | **50.59** | **52.71** |
| SAILViT | 75.42 | 53.85 | **52.02** | 54.53 |
| GranViT (SAILViT) | **76.79** | **55.40** | 51.95 | **56.61** |

Table 5: Ablation Study of the coefficient in self-distillation.

| $\lambda$ | $\alpha$ | Fine-Grained | VQA | Reasoning | OCR |
|---|---|---|---|---|---|
| 1 | 0.9 | 75.55 | 53.90 | 50.32 | 53.02 |
| 1 | 0.99 | 75.25 | 53.45 | 50.89 | 53.30 |
| 1 | 0.999 | 74.86 | 53.53 | 50.13 | 52.31 |
| 0.01 | 0.9 | 74.81 | 53.47 | 50.13 | 52.80 |
| 0.1 | 0.9 | 74.75 | 53.48 | 50.58 | 53.16 |
| 0.5 | 0.9 | 75.02 | 53.80 | 51.65 | 53.32 |

distillation training further improves fine-grained and OCR evaluations. With stage 2 adaptation, fine-grained and OCR evaluations yield additional gains of 1.0 and 0.7, respectively.

**Different Initialization for Vision Encoder.** In Table 4, we compare the performance of vision encoders initialized with different models. All three distinct vision encoders (InternViT-300M (Chen et al., 2024c), AIMv2 (Fini et al., 2025), and SAILViT-Huge (Yin et al., 2025)) exhibit significant performance improvements after pre-training, with the most notable improvements in fine-grained perception (*i.e.*, 5.3 for InternViT, 4.8 for AIMv2, and 1.3 for SAILViT) and OCR understanding (*i.e.*, 5.1 for InternViT, 4.5 for AIMv2, and 2.1 for SAILViT).

**Coefficient in Self-Distillation** We ablate two parameters in the self-distillation process: $\lambda$ and $\alpha$. To efficiently conduct the ablation experiments, during the pre-training stage, we only train stage 1 and then directly proceed to downstream SFT. As shown in Table 5, the evaluation performance of the model on fine-grained tasks gradually improves as $\lambda$ increases and $\alpha$ decreases. Therefore, we set $\lambda$ to 1 and $\alpha$ to 0.9 in our experiments.

**Visualization.** To illustrate the fine-grained feature extraction capability of GranViT, we visualize in Fig. 1(b) the attention maps of different vision encoders. AIMv2 (Fini et al., 2025) and SAIL-lViT (Yin et al., 2025) focus on global features and are severely deficient in local regions. SigLip2 (Tschannen et al., 2025) emphasizes local regions, but exhibits redundant attention to global features. In contrast, GranViT can simultaneously consider local regions and exclude interference from redundant features. This further validates the effectiveness of the proposed pre-training framework.

# 6 CONCLUSION

This paper proposed GranViT, a novel visual Transformer architecture that integrates fine-grained perception and multimodal alignment for advanced multimodal understanding. GranViT is trained on *Gran-29M*, a newly curated large-scale dataset containing global and region-level descriptive annotations for both natural and OCR images. The region-level bounding box and text annotations enable two dedicated tasks, *i.e.*, $Bbox2Caption$ for optimizing the vision encoder to strengthen fine-grained feature extraction and $Caption2Bbox$ for adapting vision features to different LLMs with enhanced region localization. Self-distillation loss is further incorporated to explicitly enhance local feature learning. GranViT is potential to serve as a robust foundation MLLM model that offers strong capabilities for complex multimodal reasoning tasks.

## 7 ACKNOWLEDGEMENT

This work was supported in part by the National Natural Science Foundation of China under Grant 62431017, Grant 62320106003, Grant 62125109, Grant 62371288, Grant 62120106007, in part by the National Key Research and Development Program of China under Grant 2025YFF0515602, and in part by the Fundamental and Interdisciplinary Disciplines Breakthrough Plan of the Ministry of Education of China uner Grant JYB2025XDXM611.

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

# A    APPENDIX

## A.1    DECLARATION ON THE USE OF LARGE LANGUAGE MODELS

In preparing this paper, we used large language models (LLMs) solely to improve the clarity of the writing. The core motivation, method design, and experimental setup are conceived and developed entirely by ourselves, without the involvement of LLMs. In addition, during the construction of the Gran-29M dataset, we employ Qwen2.5-7B for recaptioning image descriptions to improve textual quality. All final writing, editing, and formatting of the manuscript are carefully reviewed and completed by us.

## A.2    ETHICS STATEMENT

We hereby confirm that the data collection, model development, and experimental methodologies presented in this work adhere to the ICLR Code of Ethics. The Gran-29M dataset is constructed exclusively from publicly available sources and does not contain personally identifiable information or offensive content. The proposed GranViT model is designed for general-purpose visual-language understanding, and we have conducted thorough analyses to identify and mitigate potential biases in both training data and model outputs. All experiments are performed in accordance with responsible research practices, and the model will be released for research use only to prevent potential misuse. We acknowledge our responsibility to uphold ethical standards in all aspects of this research.

Table 6: Detailed data sources of datasets used in *Gran-29M*.

| Data Type | Data Source | #images | #regions |
|---|---|---|---|
| Natural | CC3M | 565521 | 2342622 |
| | IN21k | 614367 | 1628363 |
| | LAION | 17194230 | 54356988 |
| | SBU | 21479 | 52259 |
| | CC12M | 4909682 | 21714139 |
| | FLICKR30k | 1269 | 4351 |
| | YFCC15M | 655400 | 1884172 |
| | VisualGenome | 2150 | 14825 |
| OCR | Arxiv | 2655630 | 22574019 |
| | InfoVQA | 4343 | 21129 |
| | LRV-Instruction | 8304 | 22152 |
| | OCRVQA | 86 | 238 |
| | PDFVQA | 12253 | 98783 |
| | POIE | 898 | 12881 |
| | SROIE | 994 | 25586 |
| | PubTables_en | 121330 | 522516 |
| | RenderedText | 7031 | 20003 |
| | MMC-Instruction | 58583 | 218288 |
| | AI2D_gpt4v | 1958 | 27919 |
| | AI2D_internvl | 11981 | 110438 |
| | ArxivQA | 52517 | 1532997 |
| | Chart2Text | 22051 | 698874 |
| | Diagram_Image_To_Text | 154 | 2244 |
| | Robut_SQA | 5714 | 740443 |
| | Robut_WikiSQL | 38935 | 7092396 |
| | Docx_en | 429182 | 3720371 |
| | AI2D_original | 2364 | 27348 |
| | FigureQA | 96000 | 1316177 |
| | Hitab | 2495 | 392662 |
| | Robut_wtq | 38241 | 5677381 |
| | TextCaps | 20548 | 186867 |
| | TextOCR | 23511 | 215809 |
| | Uber_Text | 118042 | 571779 |
| | CORD | 955 | 3482 |
| | ChartQA | 14650 | 45871 |
| | DocBank | 25482 | 172933 |
| | SynthText | 756552 | 3563689 |
| | Docmatrix | 1019766 | 51945528 |

## A.3 REPRODUCIBILITY STATEMENT

We have taken several steps to ensure the reproducibility of this work. All datasets used in our experiments are publicly available, with their sources and processing methods detailed in Section 4. Model architectures and initializations are based on publicly released visual encoders and large language models, as described in Section 5. All training hyperparameters, including optimizer settings and learning rate schedules, are explicitly provided in Section 5.2. Code implementations and configuration files will be made publicly available upon acceptance to further facilitate replication of our results.

## A.4 DETAILS ABOUT *Gran-29M*

We systematically document the data sources of both natural and OCR images utilized in the *Gran-29M* dataset, as shown in Table 6 and Table 7. For natural images, CC3M (Sharma et al., 2018), IN21k (Deng et al., 2009), SBU (Ordonez et al., 2011), CC12M (Changpinyo et al., 2021), YFCC15M (Kamath et al., 2021), VisualGenome (Krishna et al., 2017), together with LAION (Schuhmann et al., 2022) and FLICKR30k (Young et al., 2014), are contained. For OCR im-

Table 7: Data sources of natural and OCR images in *Gran-29M*. $\#images$ and $\#regions$ denote the number of images and annotated bounding boxes after filtering, respectively.

| Data Type | Data Source | $\#images$ | $\#regions$ |
|---|---|---|---|
| Natural | UMG-41M | 6.7M | 27.63M |
| | LAION | 17.19M | 54.35M |
| | FLICKR30k | 1269 | 4351 |
| OCR | Text Images | 1.9M | 42.57M |
| | Chart, Table | 325K | 2.8M |
| | Invoice Receipt | 2847 | 41K |
| | Rich Text Images | 3.3M | 56M |
| | TOTAL | 29.51M | 183.55M |

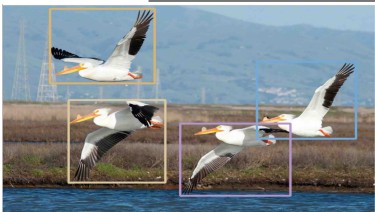

**Global Caption**
The image captures a moment of flight featuring four birds. These birds, with their predominantly white plumage accenteduated by black and grey markings, are mid-flight above what appears to be a body of water. Their wings are spread wide, showcasing the impressive span that these birds have. The be visible details suggest they may be pelicans, known for their large size and distinctive bill shape. In the background, there's a landscape with some structures that resemble towers or transmission lines, suggesting this might be near a human settlement or infrastructure.

**Local Region**

[546, 141, 759, 334], Caption: The bird in the lead.
[381, 296, 618, 477], Caption: A white bird with an orange beak.
[141, 237, 349, 444], Caption: A bird flying with its wings up.
[101, 8, 326, 195], Caption: A bird flying above three other birds

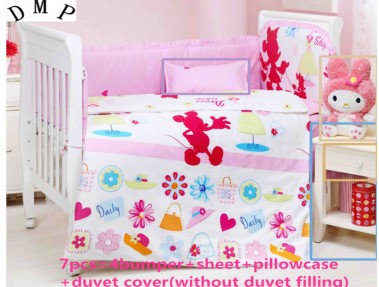

**Global Caption**
The image captures a charmingly decorated children's bedroom. Dominating is a white crib, adorned with a bedsheet and pillowcase featuring playful cartoon characters in vibrant colors. The text overlay on the image indicates that this bedding set includes 7 pieces: a bumper sheet, pillowcase, and a duvet cover (without filling. To the right of the crib, a plush pink rabbit toy sits comfortably, adding to the room's child-friendly ambiance. On the opposite side, a red cup is visible, perhaps indicating for an adult or older sibling who might be taking care of the young child. The overall setting suggests careful consideration of color coordination and character themes, creating a welcoming environment for a child.

**Local Region**

[558, 268, 679, 558], Caption: A nightstand with a white top and red bottom.
[287, 131, 445, 205], Caption: A pink pillow on the bed.
[575, 125, 679, 304], Caption: A pink stuffed animal with a bow on its head.
[582, 312, 679, 389], Caption: The drawer is white.

Figure 6: Visualization of *Gran-29M*.

ages, there are 30 datasets contained in toal for diversity: Arxiv, InfoVQA (Mathew et al., 2022), LRV-Instruction (Liu et al., 2023a), OCRVQA (Mishra et al., 2019), PDFVQA (Ding et al., 2023), POIE (Kuang et al., 2023), SROIE (Huang et al., 2019), PubTables_en (Smock et al., 2022), RenderedText (Wendler, 2024), MMC-Instruction (Liu et al., 2024a), AI2D_gpt4v (Li et al., 2024a), AI2D_internvl (Chen et al., 2024c), ArxivQA (Li et al., 2024b), Chart2Text (Kantharaj et al., 2022), Diagram_Image_To_Text, Robut_SQA (Zhao et al., 2023), Robut_WikiSQL (Zhao et al., 2023), Docx_en, AI2D_original (Kembhavi et al., 2016), FigureQA (Kahou et al., 2018), Hitab (Cheng et al., 2022), Robut_wtq (Zhao et al., 2023), TextCaps (Sidorov et al., 2020), TextOCR, Uber_Text, CORD (Park et al., 2019), ChartQA (Masry et al., 2022), DocBank (Li et al., 2020), SynthText (Gupta et al., 2016) and Docmatrix (Laurençon et al., 2024). Fig 6 and Fig 7 visualize some data samples in Gran-29M.

## A.5 HIGH RESOLUTION PERFORMANCE OF GRANVIT

Additionally, we provide the evaluation results of GranViT with an image tiling strategy in the pretraining. According to image tiling(Chen et al., 2024c; Gu et al., 2024; Wang et al., 2024; Lu et al., 2024a; Team et al., 2025b), images are firstly converted into N $512 \times 512$ local patches and one global patch. All patches are simultaneously fed into the vision encoder for feature extraction. With a patch size of 16, we obtain $N \times 1024$ visual patch features. We use pixel shuffle (Gu et al., 2024; Wang et al., 2024)to compress these visual features to $N \times 256$ patches. These visual features are then recombined based on positions and fed into the projector and LLM for understanding. The evaluation results are reported in Table 8.

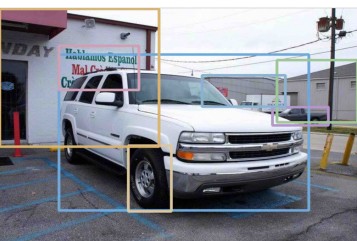

**Global Caption**
The image captures a white Chevrolet SUV parked in front of a building. The sign on the wall behind it says "Hablamosos Espanol Mal Comprendemos" which is Spanish for "Let's speak Spanish Well Understand Us". This indicates that the business may be bilingual or catering to Spanish-speaking customers.

**Local Region**
[106, 91, 553, 399], Caption: A white chevrolet truck.
[488, 195, 588, 231], Caption: A black pickup truck.
[230, 271, 307, 401], Caption: A black tire on a white truck.
[0, 1, 285, 273], Caption: A white building with a sign that says'sund.
[496, 102, 640, 225], Caption: A white house in the background.
[106, 74, 249, 161], Caption: Green and red sign.
[362, 132, 511, 194], Caption: A white truck parked in.

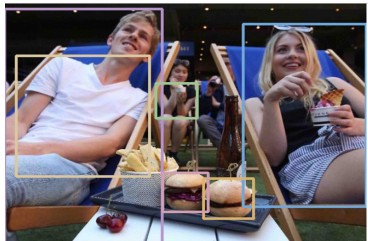

**Global Caption**
The image captures a casual and relaxed outdoor gathering. Two individuals are seated in deck chairs, with the man on the left and the woman on the right. The man is wearing a white t-shirt and jeans, while the woman is dressed in a black top and striped shorts. In front of them, there's a tray holding two burgers, one with purple toppings that may be pickles or onion relish, and another with red condiment that could be ketchup or tomato sauce. Accompanying the burgers, there's a basket filled with breadsticks and a single-use bottle of what appears to be beer. The background shows other more people sitting, suggesting a social event.

**Local Region**
[415, 39, 629, 390], Caption: A woman in a black shirt.
[0, 10, 274, 470], Caption: A man in a white shirt.
[20, 100, 252, 332], Caption: The shirt is white.
[346, 336, 433, 413], Caption: A small sandwich with purple filling.
[263, 153, 335, 220], Caption: A person sitting in the background.
[275, 325, 354, 400], Caption: The sandwich has purple sauce

Figure 7: Visualization of *Gran-29M*.

### A.6 THE REASON FOR FREEZING THE VISION ENCODER IN STAGE 2

In Table 9, we compare the performance gap between the vision encoder that is frozen and tunable in stage 2. Tuning the vision encoder in stage 2 does not achieve significant improvement, while leading to more training cost, since the vision encoder is trained well in stage 1 for fine-grained feature extraction. Therefore, to reduce the training complexity, we freeze the vision encoder in stage 2.

### A.7 THE DIFFERENCE BETWEEN GRANVIT AND SAILVIT

SAILViT (Yin et al., 2025) addresses the problem of insufficient visual-language alignment through a three-stage pre-training strategy that co-optimizes the vision encoder, projector, and LLM to achieve better alignment. GranViT differs from SAILViT in two key aspects. First, GranViT leverages both $Bbox2Caption$ and $Caption2Bbox$ tasks to strengthen fine-grained feature extraction and local region localization in the vision encoder and LLM, while SAILViT relies solely on global question-answering in the pretraining. Second, while SAILViT injects world knowledge into the vision encoder using large-scale SFT data to improve task-specific performance, GranViT focuses on enhancing the generic representation ability of the vision encoder. We contend that improved generic representations facilitate better adaptation to downstream tasks. Notably, the two strategies are complementary: after learning stronger generic features, task-specific adaptation following the paradigm of SAILViT can further enhance the performance of MLLM on specialized applications.

### A.8 CONTINUE PRETRAINING

We augment our two-stage training paradigm via continuous pretraining with SFT data, following stage 3 in SAILViT (Yin et al., 2025). We leverage LLaVA-One-Vison (Li et al., 2024a) dataset for pretraining and utilize Open-LLaVA-NeXT 1M dataset (Chen & Xing, 2024) for SFT. GranViT is initialized from SigLip2 and pretrained with *Gran-29M* dataset (8M samples for both stages) first. Then, we apply continuous pretraining with SFT data to both models. As shown in Table 11, GranViT outperforms SigLiP2 on fine-grained and OCR evaluation significantly. This experiment demonstrates that our method is compatible with the pretraining approach of SAILViT and that SFT data can be subsequently incorporated after our pretraining paradigm to further performance improvement.

Table 8: Performance comparison with image tiling. The bold font represents the best performance, and the underline represents the second performance.

| Capability | Benchmark | CLIP | SigLip | SigLip2 | AIMv2 | SAILViT | GranViT |
|---|---|---|---|---|---|---|---|
| Fine-Grained | RefCOCO_testA | 82.03 | 82.58 | 84.95 | 84.58 | 87.92 | **90.71** |
| | RefCOCO_testB | 66.53 | 67.47 | 74.91 | 70.02 | 76.85 | **82.04** |
| | RefCOCO_val | 76.01 | 77.25 | 80.11 | 77.97 | 83.12 | **87.21** |
| | RefCOCO+_testA | 75.16 | 77.68 | 80.24 | 79.23 | 82.48 | **85.15** |
| | RefCOCO+_testB | 32.00 | 55.67 | 63.81 | 59.97 | 67.13 | **70.52** |
| | RefCOCO+_val | 65.74 | 68.32 | 72.26 | 70.71 | 75.59 | **79.05** |
| | RefCOCOg_val | 70.73 | 72.74 | 75.23 | 74.91 | 78.92 | **81.98** |
| | RefCOCOg_test | 70.81 | 72.18 | 74.75 | 74.48 | 77.98 | **81.63** |
| | BLINK* | 51.40 | 52.84 | 53.49 | **56.25** | 52.33 | 52.42 |
| | MMVP | 61.33 | 64.00 | 65.33 | 67.33 | **68.00** | 65.66 |
| | Average | 67.31 | 69.07 | 72.51 | 71.55 | 75.03 | **77.64** |
| VQA | MMBench | 60.44 | 64.39 | 64.62 | **65.01** | 64.00 | 63.77 |
| | MMStar | 39.06 | 41.26 | 42.73 | 41.80 | **45.60** | 40.80 |
| | HallusionBench | 30.18 | 29.89 | 30.08 | 26.95 | 30.56 | **35.93** |
| | GQA | 58.99 | 59.88 | 59.95 | 60.04 | 60.72 | **61.32** |
| | SEEDBench | 67.35 | 68.75 | 69.60 | 69.52 | **70.07** | 69.54 |
| | Average | 51.20 | 52.83 | 53.40 | 52.66 | 54.19 | **54.27** |
| Reasoning | MMMU | 38.11 | 35.44 | **41.33** | 40.66 | 40.44 | 35.66 |
| | MathVista MINI | 36.80 | 35.80 | 37.40 | 38.90 | 38.50 | **40.00** |
| | MMVet | 35.27 | 32.47 | **40.59** | **40.59** | 40.13 | 38.30 |
| | ScienceQA | 66.18 | 68.66 | 68.36 | 66.98 | **74.71** | 66.28 |
| | AI2D | 67.13 | 68.65 | 69.62 | 69.52 | **71.85** | 69.88 |
| | Average | 48.70 | 48.20 | 51.46 | 51.33 | **53.13** | 50.02 |
| OCR | OCRBench | 414 | 450 | 545 | 522 | 551 | **583** |
| | DocVQA | 53.16 | 58.36 | 68.74 | 64.58 | 71.81 | **72.81** |
| | ChartQA | 60.12 | 62.60 | 65.04 | 65.48 | 67.36 | **71.96** |
| | InfoVQA | 24.17 | 27.81 | 31.71 | 31.38 | 33.47 | **33.59** |
| | TextVQA | 56.74 | 60.95 | 67.78 | 66.33 | **69.47** | 69.40 |
| | Average | 47.12 | 50.94 | 57.55 | 55.99 | 59.44 | **61.21** |

Table 9: Performance comparison of whether the vision encoder is frozen in stage 2.

| Vision Encoder State | FLOPs | MACs | Fine-Grained | VQA | Reasoning | OCR |
|---|---|---|---|---|---|---|
| Frozen | 3.24T | 1.62T | 77.24 | 54.83 | 51.34 | 54.02 |
| Tunable | 4.25T | 2.09T | 77.17 | 54.83 | 52.48 | 54.06 |

Table 10: Performance comparison when the vision encoder is frozen during SFT training.

| Vision Encoders | Fine-Grained | VQA | Reasoning | OCR |
|---|---|---|---|---|
| SigLip2 | 70.51 | 53.36 | 51.41 | 49.16 |
| AIMv2 | 57.31 | 52.24 | 48.75 | 46.90 |
| SAILViT | 71.90 | 54.16 | 50.93 | 52.79 |
| GranViT | 75.16 | 53.07 | 49.12 | 54.81 |

## A.9 FROZEN VISION ENCODER IN SFT

To isolate the training gains of the vision encoder during the SFT stage, we compared the performance of different vision encoders with a frozen MLLM. As shown in Table 10, GranViT achieved

Table 11: Continue training performance of GranViT. Ref, Ref+ and Refg denotes the Ref-COCO_testA, RefCOCO+_testA and RefCOCOg_test respectively. MMB, HB, and SB denote the MMBench, HallusionBench, and SEEDBench. SQA, OB, DVQA, and IVQA denote ScienceQA, OCRBench, DocVQA, and InfoVQA, respectively.

| Model | Ref | Ref+ | Refg | MMB | HB | SB | MMMU | SQA | OB | DVQA | IVQA | Avg |
|---|---|---|---|---|---|---|---|---|---|---|---|---|
| SigLip2 | 88.43 | 83.26 | 79.54 | 62.61 | 32.54 | 69.77 | 38.66 | 71.93 | 584 | 60.56 | 26.44 | 61.10 |
| GranViT | **91.26** | **86.67** | **84.32** | 61.37 | 32.44 | **70.48** | 36.77 | 70.59 | **623** | **68.16** | **28.61** | **62.99** |

the best performance in both fine-grained perception and OCR evaluations, outperforming SailViT by an average of 3.2 and 2.1, respectively. This fully demonstrates the significantly stronger fine-grained perception capability of GranViT.

## A.10 ATTENTION MAP VISUALIZATION

Similar to Fig. 1(b), we provide additional visualizations of attention maps in complex multi-object and OCR text scenarios in Fig. 8 and Fig. 9. Furthermore, to enhance the clarity of the attention maps, we filter out pixels with attention values below a threshold of 0.3, which helps eliminate diffuse activations and accentuate the model's primary focus areas. The results demonstrate that GranViT tends to concentrate more on local regions, while SailViT, SigLIP2, and AIMv2 exhibit a stronger focus on global areas.

## A.11 $Bbox2Caption$ VISUALIZATION

In Fig. 10, we provide several visualizations for the $Bbox2Caption$ task and compare the answers between GranViT and SigLip2. The red rectangle box visualizes the bbox coordinates mentioned in the questions. Unlike SigLIP2, which is limited to global image-level captioning, GranViT supports fine-grained description generation for objects inside specified bboxes.

## A.12 $Caption2Bbox$ VISUALIZATION

Fig. 11 and Fig. 12 visualizes the bbox prediction results of the $Caption2Bbox$ task in two-stage training. Green bboxes indicate predicted regions, while red ones denote ground truth. It is evident that training the $Caption2Bbox$ task in Stage 1 yields inferior performance compared to training it in Stage 2.

## A.13 BAD CASE VISUALIZATION

we visualize some failure cases in Fig. 13 in the revised manuscript. GranViT still exhibits sub-optimal performance in three challenging scenarios (from top to bottom): 1) when the bounding boxes are excessively small or text is overly dense, 2) when objects severely overlap, and 3) when precise localization is hindered by its relative coordinates. To address these limitations, we plan to explore several directions in future work. First, we will investigate the use of absolute coordinates (similar to Qwen3-VL) to improve spatial precision. Second, we will develop multi-scale pretraining strategies to enhance performance on extremely small objects. Third, we plan to incorporate advanced data augmentation techniques specifically designed for dense and overlapping scenarios. These improvements will help build a more robust and accurate visual grounding system.

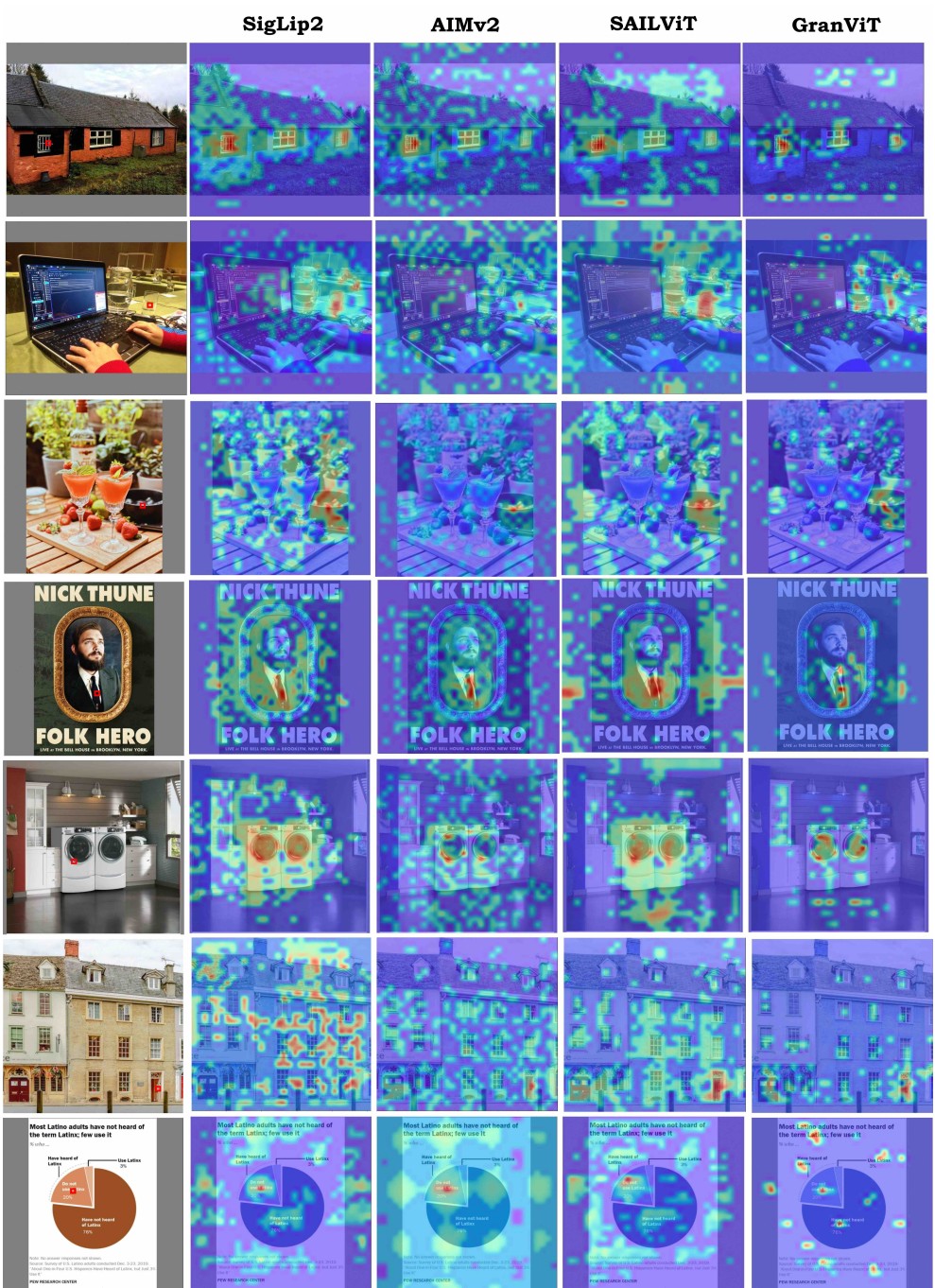

Figure 8: Attention map visualization. The red rectangle box denotes the query region for attention maps.

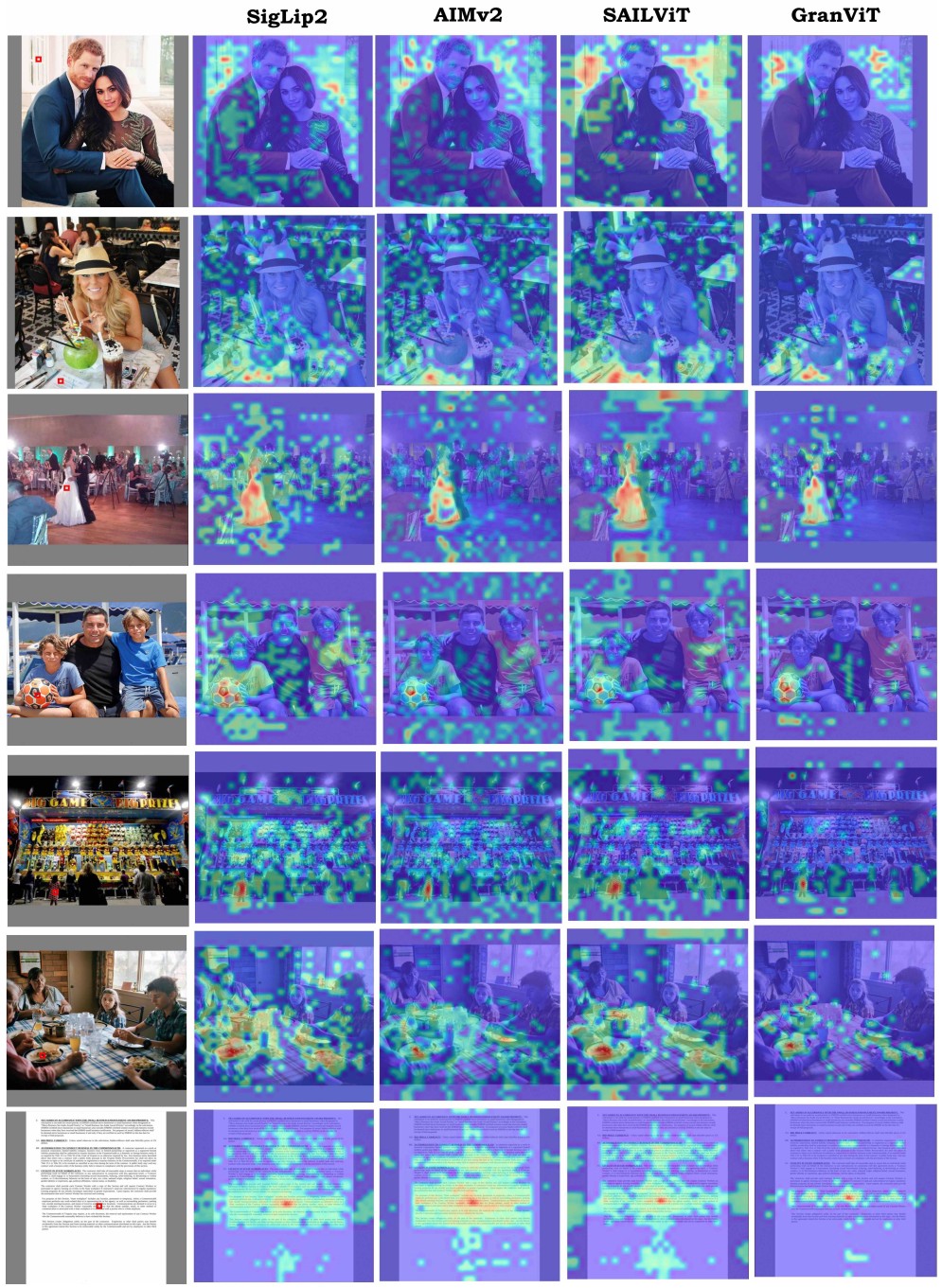

Figure 9: Attention map visualization. The red rectangle box denotes the query region for attention maps.

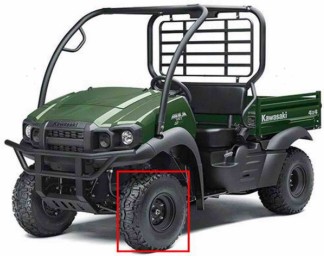

**Q: Describe the content contained within the normalized bounding box coordinates [0.366, 0.627, 0.575, 0.873] in no more than 10 words.**

SigLip2: The 2019 KAWAII Mule is a utility vehicle with an engine displacement of 450cc.

SAILViT : A green off-road vehicle

GranViT : The tire is black

Ground Truth: The tire is black

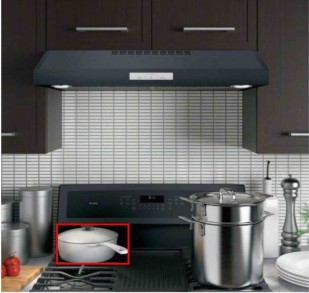

**Q: Describe the content contained within the normalized bounding box coordinates [0.19, 0.756, 0.417, 0.888] in no more than 10 words.**

SigLip2: A kitchen with a black vent hood above the stove and pots on the counter

SAILViT: This kitchen scene suggests a space ready for cooking, featuring a prominent black vent hood over the stove. Several pots sit on the adjacent counter, adding a touch of everyday life and practicality to the space.

GranViT: A white pot on the stove

Ground Truth: A white bowl on a stove

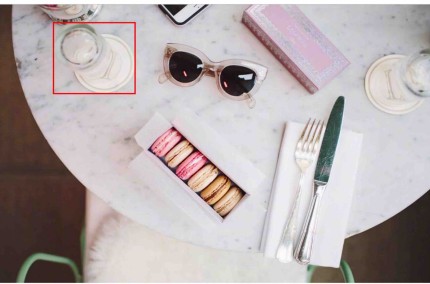

**Q: Describe the content contained within the normalized bounding box coordinates [0.138, 0.208, 0.326, 0.38] in no more than 10 words.**

SigLip2: A table with a marble top and pink glasses, forks, knife, napkins, and macarons on it.

SAILViT : An elegantly set table with a sleek marble top is prepared for a gathering. The setting includes pink-tinted drinking glasses, polished silverware, neatly folded napkins, and a delightful selection of macarons, awaiting guests.

GranViT : A glass of water

Ground Truth: A glass of water with a white label

Figure 10: *Bbox2Caption* visualization. The red rectangle box visualizes the bbox coordinates mentioned in the questions.

**Q: Please provide the bounding box coordinate of the region this sentence describes: ['a wooden man with a female partner']**

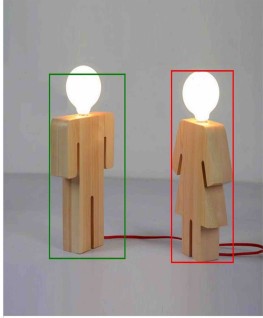

Train caption2bbox in stage1

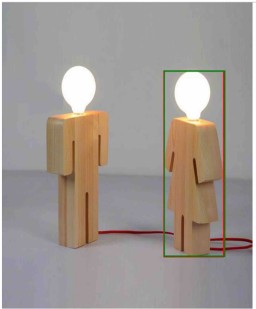

Train caption2bbox in stage2

**Q: Please provide the bounding box coordinate of the region this sentence describes: [' a wicker basket with a pillow in']**

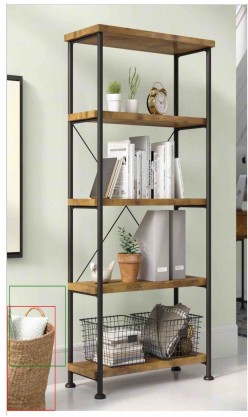

Train caption2bbox in stage1

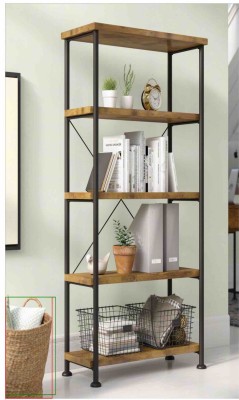

Train caption2bbox in stage2

**Q: Please provide the bounding box coordinate of the region this sentence describes: Mean age(yrs)'**

|  | Group I | Group II | Statistic | P value | NS |
|---|---|---|---|---|---|
| Number of patients | 24 | 24 |  |  |  |
| Male | 15 | 11 | X2 = 1.343 | 0.247 | * |
| Female | 9 | 13 |  |  |  |
| Mean age(yrs) | 52.31 (range,22-83) | 54.55 (range,25-78) | t = 0.448 | 0.657 | * |
| Operation side |  |  | X2 = 2.116 | 0.146 | * |
| Left | 11 | 16 |  |  |  |
| Right | 13 | 8 |  |  |  |
| Aetiology of indications |  |  | X2 = 1.532 | 0.655 | * |
| Femoral head necrosis | 16 | 18 |  |  |  |
| Femoral neck fracture | 3 | 4 |  |  |  |
| Primary osteoarthritis | 5 | 2 |  |  |  |

|  | Group I | Group II | Statistic | P value | NS |
|---|---|---|---|---|---|
| Number of patients | 24 | 24 |  |  |  |
| Male | 15 | 11 | X2 = 1.343 | 0.247 | * |
| Female | 9 | 13 |  |  |  |
| Mean age(yrs) | 52.31 (range,22-83) | 54.55 (range,25-78) | t = 0.448 | 0.657 | * |
| Operation side |  |  | X2 = 2.116 | 0.146 | * |
| Left | 11 | 16 |  |  |  |
| Right | 13 | 8 |  |  |  |
| Aetiology of indications |  |  | X2 = 1.532 | 0.655 | * |
| Femoral head necrosis | 16 | 18 |  |  |  |
| Femoral neck fracture | 3 | 4 |  |  |  |
| Primary osteoarthritis | 5 | 2 |  |  |  |

Train caption2bbox in stage1          Train caption2bbox in stage2

Figure 11: $Caption2Bbox$ visualization. Green bboxes indicate predicted regions, while red ones denote ground truth.

**Q: Please provide the bounding box coordinate of the region this sentence describes: ['flowers on the table']**

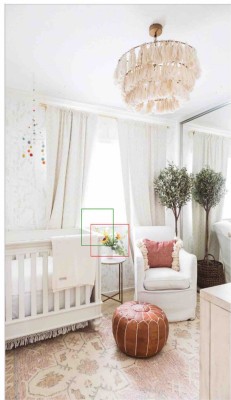 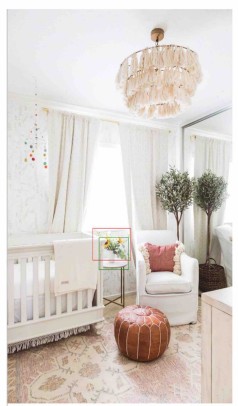

Train caption2bbox in stage1      Train caption2bbox in stage2

**Q: Please provide the bounding box coordinate of the region this sentence describes: ['a picture on the wall']**

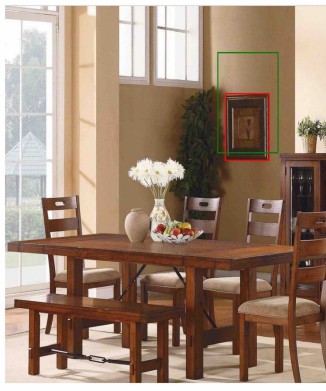 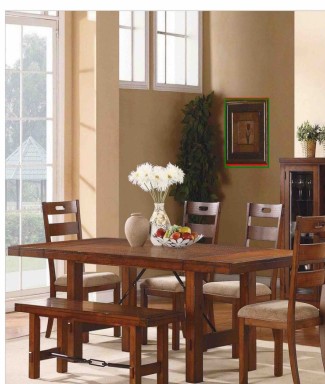

Train caption2bbox in stage1      Train caption2bbox in stage2

**Q: Please provide the bounding box coordinate of the region this sentence describes: ['the shirt is gray']**

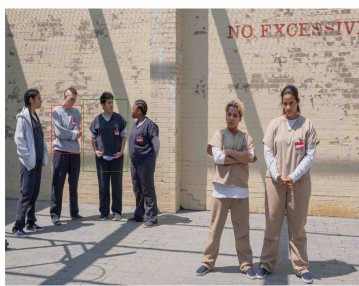 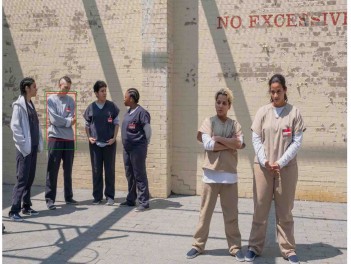

Train caption2bbox in stage1      Train caption2bbox in stage2

Figure 12: $Caption2Bbox$ visualization. Green bboxes indicate predicted regions, while red ones denote ground truth.

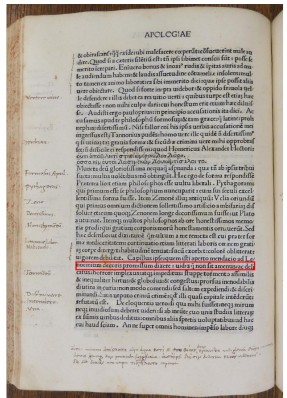

**Q: Describe the content contained within the normalized bounding box coordinates [0.334, 0.651, 0.762, 0.673] in no more than 10 words.**

GranViT: &inequaliturturhuis globosacconegtusdelfilpexpanduldeidclini

Ground Truth: nocinium decoris promffum dixcre:uidesg non fit amenusac deli

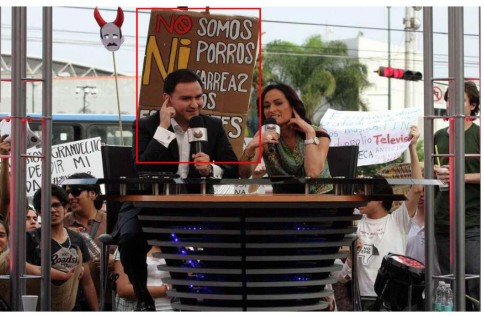

**Q: Describe the content contained within the normalized bounding box coordinates [0.288, 0.171, 0.545, 0.509] in no more than 10 words.**

GranViT: a man in a suit holding a microphone

Ground Truth: a sign in spanish

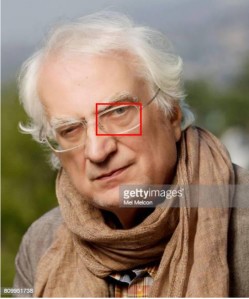

**Q: Describe the content contained within the normalized bounding box coordinates [0.41, 0.363, 0.551, 0.467] in no more than 10 words.**

GranViT: the nose of a man

Ground Truth: the eye of a man

Figure 13: Bad case visualization of $bbox2caption$ tasks. The red rectangle box visualizes the bbox coordinates mentioned in the questions. We visualize three cases (from top to bottom): 1) when the bounding boxes are excessively small or text is overly dense, 2) when objects severely overlap, and 3) when precise localization is hindered by its relative coordinates.

