# OpenReview forum: "GranViT: A Fine-Grained Vision Model For Autoregressive Multimodal Large Language Models"
_ICLR.cc/2026/Conference — ICLR 2026 Poster_

### Official Review · Reviewer_p6T8 · 2025-10-30

**Soundness:** 3
**Presentation:** 3
**Contribution:** 3
**Rating:** 6
**Confidence:** 3

**Summary:**

The paper tackles existing MLLMs’ vision encoders’ poor fine-grained perception (caused by scarce annotated data and no fine-grained pre-training). It builds the Gran-29M dataset (29M images, 180M+ region annotations) and proposes GranViT with a pretraining-adaptation framework to enhance localized representation. Experiments show GranViT outperforms other encoders, transfers well to LLMs, and hits SOTA in fine-grained recognition, VQA, and OCR tasks.

**Strengths:**

1. GranViT builds Gran-29M (29M natural/OCR images, 180M+ region annotations) to solve fine-grained data scarcity for pretraining.
2. It uses a pretraining-adaptation framework (Bbox2Caption/Caption2Bbox) plus self-distillation to boost fine-grained perception.
3. GranViT achieves SOTA in fine-grained tasks, VQA, OCR, and works well with various LLMs.

**Weaknesses:**

1. It is recommended to add the results of SAILViT in Figure 1(a).
2. In Figure 1(b), why does the visualization of SAILViT differ completely from that of other methods?
3. How was the two-stage strategy used in the paper determined, and would changing its order or combining the two stages have any impact?
4. This paper proposes Gran-29M for training, does this result in the model using far more training tokens or requiring much longer training time compared to previous methods?

**Questions:**

Refer to Weaknesses

---

> ### Author Response · Authors · 2025-11-21
>
> ## **Q1. Adding SAILViT in Figure 1**
>
> According to your suggestion, we have updated Figure 1(a) to include the results of SAILViT. To ensure a clear and fair visual comparison, we also optimized the scale of the axes. The revised figure demonstrates GranViT's outstanding performance across both fine-grained and OCR tasks compared to other methods.
>
> ## **Q2. Visualization of Attention Maps for SAILViT**
>
> We thank the reviewer's careful review for spotting this inconsistency in Figure 1(b). The discrepancy is due to an error in our visualization code, where the feature normalization is incorrectly applied to SAILViT. We have fixed this error and generated correct visualization results in Figure 1(b) in the revised manuscript. Furthermore, according to the comments raised by other reviewers, we have included more complex scenarios with multiple objects for a more comprehensive comparison, as presented in Figures 8 and 9 in the revise manuscript.
>
> ## **Q3. Ablation on Two-Stage Pretraining**
>
> We thank the reviewer for this insightful question regarding our two-stage training strategy. As discissed in Section 4.2 in the manuscript, we leverage two tasks for two pretraining stages to the enhance the vision encoder and LLM, respectively. Here, we first elaborate the role of each stage and then validate the two-stage pretraining with ablation results.
>
> In Stage 1, we freeze the LLM and train only the vision encoder and projector using the Bbox2Caption task. This focuses on producing strong visual representations. However, as shown in Figure 4(a), despite strong performance on Bbox2Caption, this task could compromise the accuracy on Caption2Bbox, since it relies more heavily on the capability of frozen LLM.
>
> In Stage 2, we freeze the pre-trained vision encoder and tune the projector and LLM. This adaptation stage specifically leverages the fine-grained features for the language model, and results in a notable accuracy gains in Caption2Bbox and marginal gains in Bbox2Caption, as shown in Figure 4(a). Figure 4(b) provides visualizations to confirm the superiority of training Caption2Bbox primarily in Stage 2.
>
> Reversing the order of two stages (i.e., Stage 1 for tuning LLM and projector and Stage 2 for training vision encoder and projector) or merging two stages into one stage could cause degraded performance. For reversing order, starting with tuning the LLM could make the vision encoder absent from foundational fine-grained training, while training all the components once is computationally expensive. Figure 4(a) shows that further optimizing Bbox2Caption in Stage 2 yields trivially benefit performance but twice the computational cost. As a result, our pretraining-adaptation paradigm that separately optimizes the vision encoder and LLM well balances the performance and efficiency.
>
> ## **Q4. Gains by Large-scale Pretraining**
>
> We thank the reviewer for pointing out this important question. We have to emphasize that, despite more time for pretraining on the Gran-29M dataset, our method enjoys the merits of rapid adaptation to varying downstream tasks and LLMs with minimal additional training. Furthermore, we validate the efficacy of GranViT pretrained using smaller scale data.
>
> **1) Pretraining for general fine-grained visual encoder.** Unlike methods like SAILViT that use 55M task-specific SFT data and could suffer from data leakage and limitd generalization, our GranViT is pretrained on our Gran-29M consisting of general data for fine-grained perception. The key efficiency gain emerges during adaptation. Once pre-trained, GranViT can be rapidly adapted to any downstream task or LLM with minimal additional training. Thus, the total cost of pre-training plus adaptation is significantly reudced, while yielding superior generalization.
>
> **2) Effectiveness on small-scale data.** We further evaluate the effectiveness of GranViT by pretraining on only 1M bbox2caption and caption2bbox samples, which equals the data scale in SFT. As reported in Table R1 below, GranViT outperforms SigLip2 in fine-grained and OCR tasks using only 1M samples for training. This further validates the effectiveness of GranViT.
>
> **Table R1: Performance comparison when transfer to Qwen2.5-14B. Ref, Ref+ and Refg denote the RefCOCO_testA, RefCOCO+_testA and RefCOCOg_test. MMB, HB, and SB stand for MMBench, HallusionBench, and SEEDBench, and. SQA, OB, DVQA, and IVQA for ScienceQA, OCRBench, DocVQA, and InfoVQA, respectively.**
>
> | Model | Ref | Ref+ | Refg | MMB | HB | SB | MMMU | SQA | OB | DVQA | IVQA |
> |-------|-----|------|------|-----|----|----|------|-----|----|------|------|
> | SigLip2 | 87.78 | 82.92 | 78.94 | 64.00 | 31.79 | 69.30 | 42.00 | 66.98 | 515 | 56.32 | 24.12 |
> | GranViT (1M) | 91.13 | 84.49 | 80.95 | 63.93 | 32.47 | 70.38 | 40.66 | 66.43 | 524 | 57.65 | 25.41 |

---

### Official Review · Reviewer_G1FV · 2025-10-31

**Soundness:** 3
**Presentation:** 3
**Contribution:** 2
**Rating:** 6
**Confidence:** 3

**Summary:**

The paper introduces a fine-grained visual representation learning framework for multimodal large language models through a two-stage training paradigm with self-distillation. The model leverages Bbox2Caption and Caption2Bbox tasks to strengthen both visual detail extraction and local region understanding, achieving strong results on OCR and fine-grained benchmarks. Experiments show that GranViT outperforms prior methods like SAILViT and SigLip2, maintaining high transferability across different LLM backbones and tasks.

**Strengths:**

1. The two-stage pretraining pipeline with both Bbox2Caption and Caption2Bbox tasks effectively enhances visual localization and detailed reasoning.
2. Table 1 results are good, demonstrating good improvements over baselines across different capabilities.
3. The experiments are comprehensive, covering relatively strong vision-language models of various sizes and benchmarks of different domains.

**Weaknesses:**

1. In Table 2, it seems that the model does not scale well with respect to the model size. Compared with the SAILViT baseline, the improvement is around 0.8 with Qwen2.5-3B and 0.4 with Qwen2.5-7B. Thus, it is unclear if the method can scale well to larger sizes.
2. From Table 1/2/3, it seems that their method doesn't lead to improvements, or even the model performance in some domains, such as MMMU and other reasoning tasks.
3. The authors claim they achieve state-of-the-art performance on certain tasks (line 109-110) while only comparing with other small-size models, which is clearly an overclaim.

**Questions:**

1. How can you expand beyond existing data sources?

---

> ### Author Response · Authors · 2025-11-21
>
> ## **Q1. Evaluation Using Larger LLM**
>
> We have to note that the apparent reduction in overall average gains (from +0.8 to +0.4) by increasing the LLM size from 3B to 7B does not indicate a scalability limitation of GranViT, since its advantages lie in fine-grained perception and OCR tasks. In fact, GranViT remains to achieve evident gains in fine-grained perception and OCR and the reduction of overall average gains is due to the reasoning tasks included in the benchmarks. GranViT obtains gains of +1.89 and +1.57 using 3B LLM, and +1.07 and +1.14 using 7B LLM in fine-grained perception and OCR, respectively. The scalability of GranViT is well validated in these tasks. For further validation, we evaluate GranViT with Qwen2.5-14B in Table.R1 below. GranViT yields sustained superior performance on fine-grained and OCR tasks and is confirmed to effectively scaled to larger models.
>
> **Table R1: Performance comparison when transfer to Qwen2.5-14B. Ref, Ref+ and Refg denote the RefCOCO_testA, RefCOCO+_testA and RefCOCOg_test. MMB, HB, and SB stand for MMBench, HallusionBench, and SEEDBench, and. SQA, OB, DVQA, and IVQA for ScienceQA, OCRBench, DocVQA, and InfoVQA, respectively.**
>
> | Model | Ref | Ref+ | Refg | MMB | HB | SB | MMMU | SQA | OB | DVQA | IVQA |
> |-------|-----|------|------|-----|----|----|------|-----|----|------|------|
> | SAILViT | 93.25 | 90.03 | 86.79 | 74.76 | 37.46 | 76.26 | 46.33 | 80.51 | 699 | 67.55 | 30.20 |
> | GranViT | 94.05 | 91.58 | 89.27 | 74.45 | 40.86 | 77.31 | 47.66 | 79.12 | 668 | 74.49 | 32.61 |
>
> ## **Q2. Explanation about Performance Gains**
>
> As discussed in Section 5.2 in the manuscript, GranViT is shown to enhance fine-grained perception and OCR capabilities, and Table 1 in the revised manuscripts shows that GranViT outperforms SAILViT by 2.83% on fine-grained benchmarks and 2.64% on OCR tasks on average. It is worth noting that GranViT (pretrained on general bbox2caption and caption2box data only) is competitive with SAILViT (pretrained on 55M task-specific intruction tuning (SFT) data and may lead to data leakage) in reasoning and VQA.
> In fact, benchmarks like MMMU rely more heavily on the LLM's inherent reasoning and knowledge integration abilities rather than fine-grained visual localization. The slight dip in reasoning performance is an expected trade-off, as pretraining in stage 2 prioritizes grounding capability, which may slightly interfere with the LLM's reasoning priors. We plan to address the limitation in future work by incorporating more reasoning-task data during SFT to better balance both capabilities.

---

> ### Author Response · Authors · 2025-11-21
>
> ## **Q3. Comparison using Larger LLMs**
>
> We evaluate the effectiveness of GranViT using larger LLMs like Qwen2.5-3B and Qwen2.5-7B in Table 2 in the revised manuscript. GranViT outperforms SAILViT by +1.89% (3B) and +1.07% (7B) in fine-grained tasks and +1.57% (3B) and +1.14% (7B) in OCR tasks. Moreover, we evaluate GranViT on LLaMA3-8B and Qwen2.5-14B in Table.R2 and Table.R3 below. GranViT consistently achieves performance gains of 1.42%, 3.37% (LLaMA3-8B) and 1.61%, 2.08% (Qwen2.5-14B) in fine-grained tasks and OCR tasks, respectively. These results validate the effectiveness of GranViT for various LLMs.
>
> **Table R2: Performance comparison for transferring vision encoders to LLama3-8B. Ref, Ref+ and Refg denote the RefCOCO_testA, RefCOCO+_testA and RefCOCOg_test. MMB, HB, and SB stand for MMBench, HallusionBench, and SEEDBench, and. SQA, OB, DVQA, and IVQA for ScienceQA, OCRBench, DocVQA, and InfoVQA, respectively.**
>
> | Model | Ref | Ref+ | Refg | MMB | HB | SB | MMMU | SQA | OB | DVQA | IVQA | Avg |
> |-------|-----|------|------|-----|----|----|------|-----|----|------|------|-----|
> | ​**Low resolution** | | | | | | | | | | | | |
> | CLIP | 90.26 | 86.43 | 80.59 | 72.07 | 37.65 | 72.18 | 48.00 | 74.90 | 462 | 44.28 | 24.14 | 61.38 |
> | SigLip | 90.47 | 86.44 | 81.44 | 74.02 | 35.25 | 74.55 | 48.97 | 76.48 | 473 | 50.69 | 23.37 | 62.63 |
> | SigLip2 | 92.85 | 88.59 | 85.59 | 72.31 | 37.99 | 74.36 | 45.62 | 76.04 | 562 | 64.72 | 27.56 | 65.62 |
> | AIMv2 | 91.30 | 88.03 | 83.67 | 72.49 | 37.02 | 73.77 | 40.77 | 73.81 | 529 | 58.85 | 24.04 | 63.33 |
> | SAILViT | 93.06 | 89.94 | 85.89 | 74.08 | 39.02 | 75.35 | 45.66 | 79.59 | 640 | 68.97 | 28.08 | 67.60 |
> | GranViT | 93.65 | 91.27 | 88.23 | 73.08 | 39.92 | 76.37 | 47.14 | 78.42 | 625 | 77.24 | 31.44 | 69.02 |
> | ​**High resolution** | | | | | | | | | | | | |
> | SigLip2 | 91.34 | 87.34 | 83.77 | 72.41 | 36.85 | 70.85 | 40.24 | 74.34 | 592 | 72.49 | 37.65 | 62.30 |
> | AIMv2 | 91.58 | 88.09 | 83.26 | 72.93 | 35.43 | 72.97 | 41.65 | 77.09 | 575 | 75.07 | 41.30 | 62.23 |
> | SAILViT | 92.57 | 90.23 | 86.58 | 75.03 | 37.62 | 72.83 | 45.29 | 80.16 | 726 | 81.30 | 43.79 | 64.11 |
> | GranViT | 93.45 | 91.35 | 88.74 | 74.85 | 37.67 | 73.06 | 44.34 | 78.89 | 702 | 82.51 | 45.67 | 64.94 |
>
> **Table R3: Performance comparison when transfer to Qwen2.5-14B. Ref, Ref+ and Refg denote the RefCOCO_testA, RefCOCO+_testA and RefCOCOg_test. MMB, HB, and SB stand for MMBench, HallusionBench, and SEEDBench, and. SQA, OB, DVQA, and IVQA for ScienceQA, OCRBench, DocVQA, and InfoVQA, respectively.**
>
> | Model | Ref | Ref+ | Refg | MMB | HB | SB | MMMU | SQA | OB | DVQA | IVQA |
> |-------|-----|------|------|-----|----|----|------|-----|----|------|------|
> | SAILViT | 93.25 | 90.03 | 86.79 | 74.76 | 37.46 | 76.26 | 46.33 | 80.51 | 699 | 67.55 | 30.20 |
> | GranViT | 94.05 | 91.58 | 89.27 | 74.45 | 40.86 | 77.31 | 47.66 | 79.12 | 668 | 74.49 | 32.61 |
>
> ## **Q4. Expanded Data Sources**
>
> Thank you for this forward-looking question. Expanding beyond existing data sources is a key direction of future work, and our data generation pipeline is well-positioned to achieve it through several strategic pathways. The core of our data creation pipeline that uses a powerful MLLM such as Qwen2.5-VL to generate regional descriptions is inherently scalable to allow for leveraging more capable models in future to produce higher-quality or novel types of annotations (e.g., video regions, 3D objects, or scientific imagery) without requiring fundamental architectural changes. Furthermore, our pretraining and adaptation paradigm is highly flexible. The GranViT encoder that is trained on a universal regional visual-language alignment task can be efficiently adapted to new modalities such as document understanding, geometric reasoning, or medical imaging through lightweight tuning on domain-specific data. We also plan to release the source codes of our data generation pipeline to encourage community collaboration in expanding the Gran-29M corpus, which will help scale data diversity and volume. In summary, our framework is not constrained by current data sources, considering its scalable model-driven data generation and modular and adaptable architectur, and our commitment to open collaboration.

---

### Official Review · Reviewer_V9VC · 2025-10-31

**Soundness:** 2
**Presentation:** 3
**Contribution:** 2
**Rating:** 4
**Confidence:** 3

**Summary:**

This paper proposes a method to enhance the fine-grained understanding capabilities of vision encoders in multimodal large language models (MLLMs). To this end, the authors first construct a dataset containing 29 million image-text pairs with fine-grained annotations, including localized bounding boxes and descriptive captions. They then introduce a two-stage training strategy that combines autoregressive learning with distillation loss to fully leverage the rich supervision provided by the dataset. Experiments on various benchmarks demonstrate the effectiveness of the proposed method and framework, achieving higher accuracy than previous approaches.

**Strengths:**

* This paper addresses an important problem and proposes an effective strategy to enhance the fine-grained understanding capabilities of vision encoders.

* The work introduces a large-scale dataset with fine-grained captions, which could be a valuable resource for the community—especially if it is open-sourced.

* The paper is well-organized and clearly written, making it easy to follow.

**Weaknesses:**

* The performance improvement over prior methods appears marginal. For example, in Table 2, the proposed method achieves only +0.83 and +0.36 average gains over SAIT-ViT on Qwen-3B and Qwen-7B, respectively. Given the relatively small margins, the practical significance and effectiveness of the proposed approach warrant further discussion.

* Qualitative examples are missing. It would be highly informative to include case studies that illustrate: (1) examples where the baseline ViT fails but GranViT succeeds, demonstrating the benefits of the proposed method; and (2) failure cases where GranViT still struggles, which would help clarify the current limitations.

* The attention visualizations in Figure 1 are not fully convincing. The images shown are relatively simple, containing only one or a few objects, and the attention maps of other ViT models also highlight local regions. To better demonstrate the advantages of GranViT, it would be more compelling to include examples from complex or text-heavy scenes (e.g., documents, charts, or cluttered environments), where precise local attention is critical.

* The impact of the hyperparameter λ is not analyzed or compared in the ablation experiments. A sensitivity study on λ would help validate the design choices and improve the reproducibility of the method.

* The title "Autoregressive Perception" may be misleading. As far as I understand, the vision encoder remains bidirectional rather than employing a causal (autoregressive) attention structure. A more descriptive title that better reflects the actual modeling approach could
improve clarity.

** Minor comments:**

There is a typo in Table 1: In the OCRBench row and SAIL-ViT column, the value "590" should likely be "59.0", as it is inconsistent with the scale of other entries.

**Questions:**

* The motivation behind the distillation loss is not fully clear. The teacher model receives a cropped image as input, while the student model processes the full image—without any textual guidance to align their feature representations. Since both are standard ViTs without task-specific conditioning, it is unclear how they are expected to produce semantically consistent features at corresponding spatial locations. Could this setup introduce ambiguity in the distillation target, and if so, how is it mitigated?

* Table 8 shows that fine-tuning the ViT leads to improved performance. However, the authors choose to keep it frozen during training due to concerns about computational cost. Given that ViTs typically have significantly fewer parameters than the LLM, it would be helpful to quantify the additional computational overhead of updating the ViT. Is the cost primarily in memory, FLOPs, or training stability? A more detailed discussion would strengthen the justification for this design choice.

* The ViT is optimized using gradients from the autoregressive loss of the LLM, which requires end-to-end training with a large language model. An alternative approach—used in several open-vocabulary object detectors such as RegionCLIP and GLIP—is to directly supervise the ViT using region-level captions in its feature space, without involving the LLM during training. This can be more computationally efficient. Could the authors discuss whether their LLM-dependent strategy offers advantages over such direct, LLM-free supervision methods, in terms of performance, scalability, or generalization?

---

> ### Author Response · Authors · 2025-11-21
>
> ## **Q1. Comparison with SAILViT**
>
> We have to emphasize that GranViT achieves evident performance gains in fine-grained and OCR tasks in comparison to SAILViT using only general bbox2caption and caption2box data. Furthermore, we demonstrate the effectiveness of our pretraining strategy by employing with GranViT maintains performance gains using the same pretraining strategy as SAILViT.
>
> **1) Superior performance using only general data.** We have elaborated in Section A.8 in the Appendix in the revised manuscripts the difference between SAILViT and GranViT in exploiting the pretraining datasets. Different from SAILViT that is pretrained on 55 million task-specific instruction-tuning (SFT) data, GranViT (initialized from SigLIP2) uses only general bbox2caption and caption2box data and may not introduce data leakage. This fact implies that GranViT's superior performance over SAILViT adequately demonstrates its effectiveness.
>
> **2) Substantial gains on fine-grained perception and OCR.** GranViT is shown to simultaneously enhance fine-grained perception and OCR and maintain the ability in VQA and reasoning. On fine-grained perception and OCR benchmarks, it consistently yields substantial average gains of +1.89% and +1.07% on fine-grained tasks and +1.57% and +1.14% on OCR tasks for the 3B and 7B models, respectively.
>
> **3) Ablation study on further pretraining strategy.** We further perform ablation study to substantiate the effectiveness of our pretraining strategy over SAILViT. We employ the same pretraining strategy as SAILViT in Table 4 in the revised manuscripts, and find consistent performance gains of our pretraining strategy to demonstrate its robustness and broader applicability.
>
> ## **Q2. Case Study Visualization**
>
> We have provided visualizations in Figures 10 in the revised manuscript to compare GranViT with SigLIP2 and SAILViT and in the bbox2caption task. Contrary to SigLIP2 and SAILViT that primarily rely on global feature extraction and yield generic scene-level captions, our GranViT excels at fine-grained localized feature understanding. These results imply that GranViT could generate precise descriptions of specific regions and effectively answer the question "what is where" with significantly greater accuracy. The comparison clearly demonstrates the fundamental difference in capability between GranViT and SigLIP2 and SAILViT.
>
> Besides, we visualize some failure cases in Figure 13 in the revised manuscript. GranViT still exhibits suboptimal performance in three challenging scenarios (from top to bottom): 1) when the bounding boxes are excessively small or text is overly dense, 2) when objects severely overlap, and 3) when precise localization is hindered by its relative coordinates. To address these limitations, we plan to explore several directions in future work. First, we will investigate the use of absolute coordinates (similar to Qwen3-VL) to improve spatial precision. Second, we will develop multi-scale pretraining strategies to enhance performance on extremely small objects. Third, we plan to incorporate advanced data augmentation techniques specifically designed for dense and overlapping scenarios. These improvements will help build a more robust and accurate visual grounding system.
>
> ## **Q3. More Attention Map Visualization**
>
> We thank the reviewer for the helpful suggestion. We agree that evaluating attention maps on more complex scenes is crucial for convincingly demonstrating GranViT's advantages. Following your advice, we have regenerated the attention map visualizations using more challenging images, including cluttered multi-object scenes and text-rich environments. Furthermore, to enhance the clarity of the attention maps, we refined our visualization code by filtering out pixels with attention values below a threshold of 0.3 to eliminate diffuse activation and accentuate the model's primary focus areas. We have presented these new results in Figure 1(b) in the revised manuscript and provided more visualizations in Figures 8 and 9 in the Appendix in the revised manuscripts.
>
> The updated visualizations robustly substantiate our claim. In complex images with multiple entities, GranViT's attention mechanism consistently and precisely localizes to the region specified by the query (the red bounding box). In contrast, models for comparison such as SAILViT, SigLip2 and AIMv2 exhibit broader, more global attention patterns. This stark difference underscores GranViT's superior capability in extracting fine-grained localized features, and suggests the critical advantage for detailed visual understanding that was not as evident in the simpler examples.

---

> ### Author Response · Authors · 2025-11-21
>
> ## **Q4. Ablation of λ**
>
> We thank the reviewer for raising this important point. To ensure clarity and reproducibility, we have indeed conducted a comprehensive sensitivity analysis. We have included it in Table 5 in the revised manuscript. It is clear that performance on fine-grained tasks improves as the weight λ on the fine-grained objective increases and the weight α on the global objective decreases. We select the configuration of λ=1 and α=0.9, since it consistently yields optimal balance to maximize the model performance. This selection of parameters provides a clear guideline for reproducibility, as suggested by the reviewer.
>
> ## **Q5. Title Change**
>
> We thank the reviewer for this insightful observation. Following the reviewer's suggestion, we will revise the title to more accurately reflect our methodological contribution. A more precise title would be: "GranViT: A Fine-Grained Vision Model For Autoregressive Multimodal Large Language Models." This modification better captures our core technical approach while maintaining clarity about the autoregressive generation process at the task level.
>
> ## **Q6. Explanation about distillation loss**
>
> We thank the reviewer for this insightful question. The core motivation of distillation loss is to introduce the explicit and direct supervision for localized features and complement the implicit pretraining from the bbox2caption task. Specifically, the teacher model receives cropped regions and is forced to develop regional features. The regional features of the student model are cropped via RoIAlign from full features. This distillation loss ensures that the distillation target (the teacher's feature) is semantically consistent with the student's regional features such that global features extracted by the student model contain region features. Since the teacher model is not fixed and is continuously updated from the EMA weight of the student model, both models evolve within a shared representation space. Therefore, rather than introducing ambiguity, this design creates a self-consistent learning paradigm where the student model is explicitly guided to produce features contained local information.
>
> ## **Q7. Comparison for Finetuning ViT**
>
> We keep the ViT frozen to yield a clear trade-off between efficiency and performance. In Table.R1 below, we compare the computation cost for freezing and fine-tuning vision encoder in stage 2. Here, we apply SigLip2 (~600M) as the vision encoder and Qwen2.5-1.5B as the LLM. We find that fine-tuning the ViT yields only marginal performance gains but incurs a significant computation overhead, i.e., an increase of approximately 1T FLOPs (23.99%) and 0.47T MACs (22.48%). This substantial increase in the total computational cost stems primarily from the need to compute and store intermediate activations throughout the ViT's deep layers during backpropagation. Therefore, frozen ViT becomes a highly efficient design choice to simultaneously maintain strong performance and greatly enhance the practicality and scalability of our method.
>
> **Table R1: Performance comparison of frozen and tunable vision encoders in Stage 2.**
>
> | Vision Encoder | FLOPs | MACs | Fine-Grained | VQA | Reasoning | OCR |
> |---------------|-------|------|--------------|-----|-----------|-----|
> | Frozen | 3.24T | 1.62T | 77.24 | 54.83 | 51.34 | 54.02 |
> | Tunable | 4.25T | 2.09T | 77.17 | 54.83 | 52.48 | 54.06 |

---

> ### Author Response · Authors · 2025-11-21
>
> ## **Q8. Comparison with RegionCLIP and GLIP**
>
> For a direct comparison, we replace our GranViT encoder with RegionCLIP and GLIP backbones, followed by SFT using Qwen2.5-1.5B. Table.R2 below shows that directly applying RegionCLIP and GLIP to MLLM architecture is degraded in downstream SFT tasks. RegionCLIP and GLIP are significantly degraded in OCR tasks due to their lack of exposure to text-rich images during pretraining. This clearly demonstrates the significant advantages of our GranViT. In addition, RegionCLIP and GLIP outperform CLIP but are significantly inferior to SAILViT and GranViT in grounding metrics (i.e. RefCOCO series) due to the feature gap. They align local features with a BERT-based textual space that differs from the semantic space of LLM, and their local features are suboptimal for grounding.
>
> **Table R2: Performance comparison with regionclip and GLIP. The best results are highlighted in bold and the second best underlined. Ref, Ref+ and Refg denote the RefCOCO_testA, RefCOCO+_testA and RefCOCOg_test. MMB, HB, and SB stand for MMBench, HallusionBench, and SEEDBench, and. SQA, OB, DVQA, and IVQA for ScienceQA, OCRBench, DocVQA, and InfoVQA, respectively.**
>
> | Model | Ref | Ref+ | Refg | MMB | HB | SB | MMMU | SQA | OB | DVQA | IVQA |
> |-------|-----|------|------|-----|----|----|------|-----|----|------|------|
> | RegionCLIP | 83.57 | 80.19 | 72.69 | 39.93 | 22.31 | 57.34 | 31.55 | 60.58 | 181 | 14.13 | 16.75 |
> | GLIP | 84.34 | 81.44 | 75.41 | 44.65 | 24.9 | 60.10 | 34.00 | 59.04 | 130 | 11.86 | 15.44 |
> | SAILViT | 89.65 | 85.01 | 80.92 | 63.54 | 31.29 | 69.75 | 38.66 | 72.78 | 590 | 58.75 | 24.75 |
> | GranViT | 91.79 | 87.04 | 83.32 | 62.46 | 30.34 | 70.34 | 38.00 | 67.42 | 551 | 67.92 | 27.19 |
>
> ## **Minor comments**
>
> We thank the reviewer for their careful reading. In Table.1 in the revised manuscripts, the value "590" for SAILViT on OCRBench is indeed correct and not a typo. The OCRBench metric uses a different scale (0-1000) compared to the other benchmarks (typically 0-100). To ensure a fair and consistent calculation of the average score across all benchmarks, we normalize the OCRBench score by dividing it by 10 when calculating the average. This aligns its scale with the other metrics.

---

> > ### Comment · Reviewer_V9VC · 2025-11-27
> >
> > I thank the authors for their detailed response. I especially appreciate the clarification regarding the distillation loss and $\lambda$, as well as the additional visualizations. However, I remain unconvinced on the following key points:
> >
> > 1. Dataset Efficiency and Generalizability: While the proposed dataset (29M) is smaller than the SAIL-ViT data (55M), the reduction in scale is not significant enough to justify its effectiveness and superiority, particularly given that the performance remains merely comparable. Since bounding-box data is crucial for learning spatial information, the trade-off presented here suggests the proposed dataset may not be as effective as claimed. Furthermore, the revised Table 5 shows only a marginal improvement (~1%) over SAIL-ViT, which raises further doubts about the method's generalizability and practical impact.
> >
> > 2. Vision Encoder Overhead (Q7): I am not persuaded by the argument for freezing the vision encoder solely to reduce computational cost. Tuning the vision encoder introduces only a ~20% overhead with a 1.5B LLM, which is generally considered acceptable in this context. Moreover, as the LLM size increases, this relative overhead diminishes further. Therefore, the necessity of freezing the encoder remains unjustified.

---

> > > ### Author Response · Authors · 2025-12-04
> > >
> > > ## Q2. Vision Encoder Overhead
> > >
> > > Tuning vision encoder produces approximately 23% overhead in computation cost on each single GPU and could cause more significant overhead in multi-node multi-GPU system. Here, we justify freezing the vision encoder by directly verifying its reduction in the training time and GPU memory usage on 64 H800 GPUs and its competitive performance, compared with tuning the vision encoder.
> > >
> > > **Training time and GPU memory usage.** We further evaluate the efficiency of freezing the vision encoder by measuring the training time and GPU memory usage in the multi-node, multi-GPU machine.
> > > We train GranViT with different LLM (Qwen2.5-1.5B, Qwen2.5-3B and Qwen2.5-7B) with 24M Caption2bbox data on 64 H800 GPUs and report the training time and peak GPU memory usage, under both low-resolution and high-resolution image settings. Table R5 consistently shows that freezing the vision encoder leads to substantial reductions in training time and GPU memory usage, which are non-negligible in practice.
> > >
> > > - High Resolution. With image tiling, freezing the vision encoder reduces the total training time by approximately 17 hours and peak GPU memory usage by 10 GB.
> > > - Low Resolution. Without image tiling, freezing the encoder still reduces the total training time by about 7 hours and memory usage by 5-10 GB.
> > >
> > > Tuning the vision encoder increases computational cost by approximately 23%, and could further significantly increases the overhead of training time and GPU memory usage in multi-GPU training. Specifically, it necessitates synchronizing gradients of the vision encoder across multiple nodes, and increases communication time and extra GPU memory for caching during backpropagation. This causes more training time and higher GPU memory usage in total.
> > >
> > > **Comparison with tuning vision encoder.**
> > > As shown in Table 9 in the revised manuscripts, tuning the vision encoder in Stage 2 does not yield significant performance gains, despite increasing total training time and GPU memory usage. Thus, freezing the encoder strikes an optimal balance between efficiency and effectiveness, particularly given the scale of our experiments.
> > >
> > > **Table R5: Stage 2 Training Efficiency Comparison (using 24M data, 64 H800 GPUs)**
> > >
> > > | Vision Encoder | Total Time | Time per Iteration | Peak GPU Memory |
> > > |---------------|------------|-------------------|-----------------|
> > > | ​**Low resolution (no image tiling)** | | | |
> > > | Tunable | 45h | 0.86it/s | 35.42GB |
> > > | Frozen | 28h | 0.53it/s | 29.35GB |
> > > | Reduction | 17h | 0.33it/s | 6.07GB |
> > > | Tunable | 55h | 1.06it/s | 54.05GB |
> > > | Frozen | 38h | 0.74it/s | 41.07GB |
> > > | Reduction | 17h | 0.32it/s | 12.98GB |
> > > | Tunable | 71h | 1.42it/s | 64.29GB |
> > > | Frozen | 56h | 1.08it/s | 53.28GB |
> > > | Reduction | 15h | 0.34it/s | 11.01GB |
> > > | ​**High resolution (image tiling)** | | | |
> > > | Tunable | 30h | 0.58it/s | 37.24GB |
> > > | Frozen | 23h | 0.45it/s | 32.02GB |
> > > | Reduction | 7h | 0.13it/s | 5.22GB |
> > > | Tunable | 39h | 0.76it/s | 56.34GB |
> > > | Frozen | 32h | 0.63it/s | 48.48GB |
> > > | Reduction | 7h | 0.33it/s | 7.86GB |
> > > | Tunable | 64h | 1.07it/s | 66.87GB |
> > > | Frozen | 47h | 0.92it/s | 56.46GB |
> > > | Reduction | 17h | 0.15it/s | 10.41GB |

---

> ### Author Response · Authors · 2025-12-04
>
> ## Q1. Dataset Efficiency and Generalizability
>
> We have to emphasize that the effectiveness of our Gran-29M dataset lies in its ability to train models for general fine-grained perception while maintaining generalization ability. Therefore, different from SAILViT that directly stems from SFT data in pretraining and could overfit in-domain tasks, our GranViT trained with Gran-29M for general fine-grained perception achieves general representation and significant generalization ability.
>
> **Evident performance gains.** GranViT along with Gran-29M have shown evident performance gains in several (fine-grained perception and OCR benchmarks and extensive 3D, GUI and realworld benchmarks, as reported in Table R3.
>
> - GranViT significantly outperforms SAILViT by 2.14% on RefCOCO_testA, 2.03% on RefCOCO+_testA, 9.21% on DocVQA, and 4.72% in ChartVQA.
> - SAILViT struggles with the perception of 3D objects, GUI components, and in-the-wild images. Even compared with SigLip2, it suffers from performance losses of 2.37% in CVBench3D, 3.39% in RealWorldQA, and 2.40% in WildVision, since SAILViT overfits to its SFT data and are limited in generalization ability. On the contrary, GranViT outperforms SAILViT by a large margin in these benchmarks, i.e., 4.67% in CVBench3D, 4.56% in RealWorldQA, and 5.8% in WildVision.
>
> **Scalability with more pretraining data.** It is worth mentioning that the results in Table 4 are obtained using GranViT pretrained with 8M Bbox2caption and Caption2bbox data. To validate the scalability of GranViT with more pretraining data, we further pretrain GranViT using 16M Bbox2caption and Caption2bbox data. Table R4 demonstrates that, with more pretraining data, the performance of GranViT can be further improved, underscoring its scalability and potential.
>
> **Table R3: Performance comparison in In-domain and Out-of-domain benchmarks. Ref, Ref+ and Refg denote the RefCOCO_testA, RefCOCO+_testA and RefCOCOg_test.DVQA, CVQA and IVQA for DocVQA, ChartVQA and InfoVQA, respectively.**
>
> | Model | Ref | Ref+ | Refg | BLINK* | DVQA | CVQA | IVQA |
> |-------|-----|------|------|--------|------|------|------|
> | ​**Fine-grained and OCR benchmarks** | | | | | | | |
> | SigLip2 | 87.78 | 82.92 | 78.94 | 50.35 | 56.32 | 64.44 | 24.12 |
> | SAILViT | 89.65 | 85.01 | 80.92 | 52.54 | 58.75 | 63.24 | 24.75 |
> | GranViT | 91.79 | 87.04 | 83.82 | 56.80 | 67.92 | 67.96 | 27.19 |
> | Improvement | 2.14 | 2.03 | 2.9 | 4.26 | 9.17 | 4.72 | 2.44 |
> | ​**Out-of-domain benchmarks** | **CVBench2D** | **CVBench3D** | **SpatialEval** | **RealWorldQA** | **HRBench4K** | **WildVision** | **ScreenSpot** |
> | SigLip2 | 65.27 | 51.53 | 32.51 | 52.54 | 32.87 | 21.80 | 12.6 |
> | SAILViT | 64.97 | 49.16 | 34.41 | 49.15 | 33.25 | 19.40 | 11.5 |
> | GranViT | 66.36 | 53.83 | 37.17 | 53.72 | 37.37 | 25.20 | 16.1 |
> | Improvement | 1.39 | 4.67 | 2.76 | 4.57 | 4.12 | 5.8 | 4.6 |
>
> **Table R4: Performance with SAILViT initialization for GranViT during pretraining.**
>
> | Model | Fine-Grained | VQA | Reasoning | OCR |
> |-------|--------------|-----|-----------|-----|
> | ​**pretrained with 8M Bbox2caption and Caption2bbox data** | | | | |
> | SAILViT | 75.42 | 53.85 | 52.02 | 54.53 |
> | GranViT (SAILViT) | 76.79 | 55.40 | 51.95 | 56.61 |
> | ​**pretrained with 16M Bbox2caption and Caption2bbox data** | | | | |
> | SAILViT | 75.42 | 53.85 | 52.02 | 54.53 |
> | GranViT (SAILViT) | 78.53 | 56.02 | 52.00 | 58.94 |

---

### Official Review · Reviewer_c9S8 · 2025-11-02

**Soundness:** 2
**Presentation:** 3
**Contribution:** 3
**Rating:** 6
**Confidence:** 4

**Summary:**

The paper proposes GranViT, a vision encoder for MLLMs aimed at fine-grained perception. The key ideas are: 1) Proposes a pretraining corpus Gran-29M of 29.5M images with 183.6M region-level annotations built by combining natural-image and OCR sources and auto-generated captions and bounding boxes. 2) Two-stage training recipe: tage-1 tunes the vision encoder + projector with LLM frozen using Bbox2Caption; Stage-2 freezes the vision encoder and tunes projector + LLM using Caption2Bbox to strengthen localization and transferability. The results show SOTA or strong performance on fine-grained detection and OCR tasks, comparable VQA, and slight trade-offs in general reasoning.

**Strengths:**

- Clear problem focus and a method explicitly tailored to address the problem. The 2-stage training procedure is well-motivated and empirically supported.
- Contribution of a large, diverse, and structured pretraining data with concrete filtering procedure.

**Weaknesses:**

- Data quality control: Gran-29M relies heavily on auto-generated annotations, it remains unclear whether the dataset quality is good enough for large-scale training. A data quality auditing process needs to be applied.
- The same Qwen2.5-VL family was used to generate captions and later to consume the vision features in adaptation stage. It remains unclear whether the model performance gain was inflated by distributional closeness to its own generated text style or by leakage.

**Questions:**

1. Is there any verification or auditing of the auto-generated data annotations? How noisy are the Qwen-generated local captions?
2. The Qwen2.5-VL models were used to generate the captions then also used to serve as the LLM backends. Did you evaluate with non-Qwen LLMs for captioning or LLM backends?
3. Localization baselines: It would be great to include comparisons against task-specific localization and grounding models, beyond the existing MLLM-encoder baselines.

---

> ### Author Response · Authors · 2025-11-21
>
> ## **W1&Q1. Data quality control**
>
> We assess the quality of Gran-29M by two critera: 1) model-based criteria and 2) human criteria.
>
> **1) Model-based criteria.** According to [1], we employ Qwen3-VL-32B for evaluation across three dimensions: QA correspondence, QA relevance, and visual dependence. Each dimension is assigned a score ranging from 1 to 5, where score 1 indicates the poorest quality and score 5 for the highest quality. Table.R1 and Table.R2 below show the distribution of scores and demonstrates that most of our auto-generated samples yield high quality across all the three dimensions, i.e., 89.0%, 83.6%, and 99.9% yielding high ratings of scores 4 or 5 on natural images and 90.6%, 87.4%, and 99.2% on OCR images, respectively.
>
> **2) Human criteria.** We further performed a human audit on a random sample of 500 data points from Gran-29M to ensure reliability. Table.R3 and Table.R4 below show consistent results for human evaluations with the model-based assessment. The ratios of generated samples with scores 4 and 5 are 89.0%, 89.8%, and 99.98% for random samples and 93.4%, 92.2%, and 90.28% for OCR images.
>
> **Table R1: Qwen3VL-32B evaluation results on natural images.**
>
> | Dimension | Score 1 | Score 2 | Score 3 | Score 4 | Score 5 |
> |-----------|---------|---------|---------|---------|---------|
> | QA Correspondence | 12306 (2.1%) | 10663 (2.5%) | 31951 (6.4%) | 122536 (24.5%) | 322517 (64.5%) |
> | QA Relevance | 13751 (2.2%) | 10821 (2.8%) | 57446 (11.5%) | 161022 (32.2%) | 256943 (51.4%) |
> | Visual Dependence | 0 (0.0%) | 502 (0.1%) | 0 (0.0%) | 504 (0.1%) | 498994 (99.8%) |
>
> **Table R2: Qwen3VL-32B evaluation results on OCR images.**
>
> | Dimension | Score 1 | Score 2 | Score 3 | Score 4 | Score 5 |
> |-----------|---------|---------|---------|---------|---------|
> | QA Correspondence | 18748 (1.5%) | 7442 (1.8%) | 13374 (3.2%) | 34672 (8.3%) | 344736 (82.3%) |
> | QA Relevance | 15131 (3.8%) | 18568 (4.6%) | 16709 (4.2%) | 73823 (18.4%) | 276769 (69.0%) |
> | Visual Dependence | 0 (0.0%) | 2516 (0.5%) | 1505 (0.3%) | 3527 (0.7%) | 492452 (98.5%) |
>
> **Table R3: Human evaluation results on natural images**
>
> | Dimension | Score 1 | Score 2 | Score 3 | Score 4 | Score 5 |
> |-----------|---------|---------|---------|---------|---------|
> | QA Correspondence | 5 (1.0%) | 8 (1.6%) | 42 (8.4%) | 185 (37.0%) | 260 (52.0%) |
> | QA Relevance | 7 (1.4%) | 9 (1.8%) | 35 (7.0%) | 198 (39.6%) | 251 (50.2%) |
> | Visual Dependence | 0 (0%) | 0 (0%) | 1 (0.02%) | 1 (0.02%) | 498 (99.96%) |
>
> **Table R4: Human evaluation results on OCR images**
>
> | Dimension | Score 1 | Score 2 | Score 3 | Score 4 | Score 5 |
> |-----------|---------|---------|---------|---------|---------|
> | QA Correspondence | 3 (0.6%) | 5 (1.0%) | 25 (5.0%) | 210 (42.0%) | 257 (51.4%) |
> | QA Relevance | 4 (0.8%) | 7 (1.4%) | 28 (5.6%) | 205 (41.0%) | 256 (51.2%) |
> | Visual Dependence | 0 (0.0%) | 6 (0.012%) | 3 (0.006%) | 40 (0.08%) | 451 (90.2%) |
>
> [1] Wiedmann, Luis, et al. "Finevision: Open data is all you need." arXiv preprint arXiv:2510.17269 (2025).

---

> ### Author Response · Authors · 2025-11-21
>
> ## **W2&Q2. Non-Qwen LLM**
>
> We thank the reviewer for this insightful question. We perform an additional ablation study to rule out the potential inflation of performance gains due to distributional closeness or feature leakage and further validate that the gains stem from the generalizability of GranViT's visual features rather than a closed-loop artifact from two perspectives.
>
> **1)** In addition to the Qwen family, we evaluate the transferability of our GranViT with Llama3-8B, a fundamentally different LLM. Table.R5 below shows that we achieve superior performance on Llama3. This robustly demonstrates that the gains are not specifically limited to Qwen's self-generated text style, since Llama3 has a distinct architecture and pre-training distribution from Qwen, and confirms the authenticity and transferability of the visual representations learned by GranViT.
>
> **2)** GranViT yields leading performance on fine-grained and OCR tasks that demand accurate pixel-level understanding. On fine-grained perception and OCR benchmarks, it consistently yields substantial average gains of +1.89% and +1.07% on fine-grained tasks and +1.57% and +1.14% on OCR tasks for the 3B and 7B models, respectively. This further supports the conclusion that the improvements are attributable to the GranViT architecture itself.
>
> **Table R5: Performance comparison for transferring vision encoders to LLama3-8B. Ref, Ref+ and Refg denote the RefCOCO_testA, RefCOCO+_testA and RefCOCOg_test. MMB, HB, and SB stand for MMBench, HallusionBench, and SEEDBench, and. SQA, OB, DVQA, and IVQA for ScienceQA, OCRBench, DocVQA, and InfoVQA, respectively.**
>
> | Model | Ref | Ref+ | Refg | MMB | HB | SB | MMMU | SQA | OB | DVQA | IVQA | Avg |
> |-------|-----|------|------|-----|----|----|------|-----|----|------|------|-----|
> | ​**Low resolution** | | | | | | | | | | | | |
> | CLIP | 90.26 | 86.43 | 80.59 | 72.07 | 37.65 | 72.18 | 48.00 | 74.90 | 462 | 44.28 | 24.14 | 61.38 |
> | SigLip | 90.47 | 86.44 | 81.44 | 74.02 | 35.25 | 74.55 | 48.97 | 76.48 | 473 | 50.69 | 23.37 | 62.63 |
> | SigLip2 | 92.85 | 88.59 | 85.59 | 72.31 | 37.99 | 74.36 | 45.62 | 76.04 | 562 | 64.72 | 27.56 | 65.62 |
> | AIMv2 | 91.30 | 88.03 | 83.67 | 72.49 | 37.02 | 73.77 | 40.77 | 73.81 | 529 | 58.85 | 24.04 | 63.33 |
> | SAILViT | 93.06 | 89.94 | 85.89 | 74.08 | 39.02 | 75.35 | 45.66 | 79.59 | 640 | 68.97 | 28.08 | 67.60 |
> | GranViT | 93.65 | 91.27 | 88.23 | 73.08 | 39.92 | 76.37 | 47.14 | 78.42 | 625 | 77.24 | 31.44 | 69.02 |
> | ​**High resolution** | | | | | | | | | | | | |
> | SigLip2 | 91.34 | 87.34 | 83.77 | 72.41 | 36.85 | 70.85 | 40.24 | 74.34 | 592 | 72.49 | 37.65 | 62.30 |
> | AIMv2 | 91.58 | 88.09 | 83.26 | 72.93 | 35.43 | 72.97 | 41.65 | 77.09 | 575 | 75.07 | 41.30 | 62.23 |
> | SAILViT | 92.57 | 90.23 | 86.58 | 75.03 | 37.62 | 72.83 | 45.29 | 80.16 | 726 | 81.30 | 43.79 | 64.11 |
> | GranViT | 93.45 | 91.35 | 88.74 | 74.85 | 37.67 | 73.06 | 44.34 | 78.89 | 702 | 82.51 | 45.67 | 64.94 |

---

> ### Author Response · Authors · 2025-11-21
>
> ## **Q3. Comparison with task-specific models**
>
> We thank the reviewer for the constructive suggestion. We have included in Table.R6 below the comparison with specialized referring expression models on RefCOCO series benchmarks. The results demonstrate that GranViT achieves comparable performance to task-specific models and surpasses them in terms of metrics such as RefCOCO+_testA, RefCOCOg_val, RefCOCOg_test. More importantly, unlike specifically designed models limited to specific tasks, GranViT is a versatile visual encoder that enables reasoning, visual question answering, and OCR, and provides a more practical and efficient solution to develop general-purpose multimodal systems with its capability to handle diverse tasks.
>
> **Table R6: Localization and grounding performance comparision between GranViT and other specialists**
>
> | Type | Method | RefCOCO_val | RefCOCO_testA | RefCOCO_testB | RefCOCO+_val | RefCOCO+_testA | RefCOCO+_testB | RefCOCOg_val | RefCOCOg_test |
> |-------|-----------------|-------------|----------------|----------------|--------------|-----------------|-----------------|--------------|---------------|
> | Specialists | MDETR | 87.5 | 90.4 | 82.7 | 81.1 | 85.5 | 73.0 | 83.3 | 83.3 |
> | Specialists | G-DINO-L | 90.6 | 93.2 | 88.2 | 82.8 | 89.0 | 75.9 | 86.1 | 87.0 |
> | Generalists | Ferret-7B | 87.5 | 91.4 | 82.5 | 80.8 | 87.4 | 73.1 | 83.9 | 84.8 |
> | Generalists | Shikra-7B | 87.0 | 90.6 | 80.2 | 81.6 | 87.4 | 72.1 | 82.3 | 82.2 |
> | Generalists | Groma | 89.5 | 92.1 | 86.3 | 83.9 | 88.9 | 98.1 | 86.4 | 87.0 |
> | Generalists | Pink | 88.7 | 92.1 | 84.0 | 81.8 | 88.2 | 73.9 | 83.9 | 84.3 |
> | Generalists | ROD-MLLM | 90.2 | 93.0 | 86.3 | 84.8 | 89.9 | 77.5 | 86.7 | 86.7 |
> | Generalists | GranViT-LLM | 91.12 | 92.98 | 87.71 | 86.15 | 90.46 | 79.99 | 87.90 | 87.96 |

---

### Meta-Review · Area_Chair_CZCY · 2026-01-06

**Summary:**

The paper proposes a ViT with enhanced fine-grained perception. The main components are pre-training data called Gran-29M (29M images with 180 million region-level annotations) and a two-stage pretraining strategy involving Bbox2Caption and Caption2Bbox tasks and self-distillation.

The paper receives ratings of 4, 6, 6, 6. The discussions revolve around 1. Quality control for Gran-29M (Reviewer c9S8), 2. Transferability of GranViT: Potential bias due to the use of the same family of Qwen in different stages (Reviewer c9S8), 3. Significance of the results in comparison to RegionCLIP, GLIP, and especially SAILViT (​Reviewer V9VC), 4. Other ablation experiments (Reviewer G1FV and p6T8): larger LLMs, smaller-scale data.

**Reviewer Concerns:**

The authors robustly present results to address the reviewers one by one. 1. Present automatic and human evals for Gran-29M. 2. Use Llama3-8B as LLM. 3. Clarify the significance over SAILViT. 4. Include experiments to further support the utility and generalizability of GranViT’s components.

**Reviewer Scores:**

The reviewers would likely keep the scores or increase them (6+, 6+, 6+, 4+). The only reviewer with the rating of 4 remained unconvinced after the authors’ first response but didn’t get to respond to another comment by the authors (especially regarding SAILViT). Overall, the concerns look sufficiently addressed.

---

### Decision · Program_Chairs · 2026-01-26

Accept (Poster)